# Production of recombinant human IgG1 Fc with beneficial N-glycosylation pattern for anti-inflammatory activity using genome-edited chickens

Jin Se Park[1,2,3], Hee Jung Choi[1,3], Kyung Min Jung [1], Kyung Youn Lee[1], Ji Hyeon Shim[1], Kyung Je Park[1], Young Min Kim[1,2] & Jae Yong Han [1✉]

Intravenous immunoglobulin (IVIG) is a plasma-derived polyclonal IgG used for treatment of autoimmune diseases. Studies show that α-2,6 sialylation of the Fc improves anti-inflammatory activity. Also, afucosylation of the Fc efficiently blocks FcγRIIIA by increasing monovalent affinity to this receptor, which can be beneficial for treatment of refractory immune thrombocytopenia (ITP). Here, we generated genome-edited chickens that synthesize human IgG1 Fc in the liver and secrete α-2,6 sialylated and low-fucosylated human IgG1 Fc (rhIgG1 Fc) into serum and egg yolk. Also, rhIgG1 Fc has higher affinity for FcγRIIIA than commercial IVIG. Thus, rhIgG1 Fc efficiently inhibits immune complex-mediated FcγRIIIA crosslinking and subsequent ADCC response. Furthermore, rhIgG1 Fc exerts anti-inflammatory activity in a passive ITP model, demonstrating chicken liver derived rhIgG1 Fc successfully recapitulated efficacy of IVIG. These results show that genome-edited chickens can be used as a production platform for rhIgG1 Fc with beneficial N-glycosylation pattern for anti-inflammatory activities.

[1] Department of Agricultural Biotechnology and Research Institute of Agriculture and Life Sciences, Seoul National University, Seoul, Republic of Korea. [2] Avinnogen Co., Ltd, 1 Gwanak-ro, Gwanak-gu, Seoul, Republic of Korea. [3] These authors contributed equally: Jin Se Park, Hee Jung Choi. ✉email: jaehan@snu.ac.kr

Autoimmune diseases (ADs) are caused by an aberrant immune response against self-antigens[1]. Generally, autoantibodies that recognize self-antigens trigger inflammatory responses by activating effector cells such as macrophages, neutrophils, and natural killer cells, resulting in the destruction of tissues or cells[2]. Therefore, to treat ADs, it is important to inhibit effector cell activation and dampen inflammatory responses. The anti-inflammatory agent intravenous immunoglobulin (IVIG) is used widely to treat inflammatory ADs[3]. When IVIG is infused at high doses (i.e., 1–2 g/kg), it can ameliorate inflammation and recover platelet counts in patients with immune thrombocytopenia (ITP)[4]. Because IVIG shows anti-inflammatory activity against numerous ADs, global demand is increasing continuously[5]. However, because IVIG is prepared from donated human plasma, supply shortages and high costs are major limitations[6]. Therefore, it is desirable to develop an efficient system for the production and supply of IVIG alternatives that recapitulate the anti-inflammatory activity of plasma-derived IVIG.

Although IVIG acts via diverse mechanisms, its Fc fragment is critical for anti-inflammatory activity[7–9]. It is known that α-2,6 sialylated Fc promotes expression of FcγRIIB in effector macrophages by promoting secretion of $T_H2$ cytokines such as IL-33 and IL-4, and the abundant ratio of α-2,6 sialylated Fc increases the anti-inflammatory activity of IVIG significantly[10–15]. Therefore, highly α-2,6 sialylated Fc is a potential IVIG alternative that shows enhanced anti-inflammatory activity.

The other major mechanism underlying the activity of IVIG is the competitive blockade of activating Fcγ receptors (FcγRs)[7,16]. When IVIG is administered at high doses, it occupies activating FcγRs and prevents immune complexes from binding and crosslinking these receptors[17,18]. In particular, the anti-inflammatory activity of IVIG is FcγRIIIA-dependent, suggesting that blocking FcγRIIIA is a potential mechanism of IVIG activity[19–21]. This notion is supported by a study showing that blockade of FcγRIIIA by monoclonal antibodies recapitulates IVIG activity and efficiently ameliorates inflammatory responses in refractory ITP patients, although FcγRIIIA crosslinking results in adverse side effects[22]. Because absence of core fucosylation (afucosylation) significantly increase the monovalent affinity of IgG to FcγRIIIA[23], it can be assumed that afucosylated Fc will efficiently occupy FcγRIIIA and will be beneficial for inducing anti-inflammatory activity and treatment of refractory ITP. Recently, it was demonstrated that FcγRIIIA of NK cells was abundantly occupied by afucosylated IgG[24]. Furthermore, it was shown that afucosylated antigen-aspecific antibodies efficiently block FcγRIIIA and can be anti-inflammatory by inhibiting Fc effector functions of antigen-specific antibody[25].

Therefore, if we can generate an hIgG1 Fc region that has a high levels of α-2,6 sialylated N-glycans and simultaneously has high affinity for FcγRIIIA via afucosylation, we can elicit the multiple modes of action of IVIG, thereby improving anti-inflammatory activity. However, plasma derived IVIG contains low amounts of sialylated Fc (around 10% of total N-glycans) and has a large proportion of fucosylated N-glycans[26]. Recombinant hIgG1 Fc derived from mammalian cell culture also has low levels of sialylated, and high levels of fucosylated glycans unless specific modifications are made to the expressing cell lines[27]. Therefore, an alternative system for the production of hIgG1 Fc containing a high proportion of α-2,6 sialylated and afucosylated N-glycans is required for development of an IVIG alternative with enhanced anti-inflammatory activity.

Chickens are an efficient platform for the production of biopharmaceuticals; this is because hen's eggs contain abundant proteins, and hens lay over 300 eggs per year[28]. Also, the glycosylation pattern of chicken egg proteins is similar to that of human proteins, and more importantly, chickens only produce human-like glycans and do not produce immunogenic α-galactose and Neu5Gc[28,29]. For these reasons, various approaches to increasing the accumulation of biopharmaceuticals in eggs, particularly egg whites, have been studied[30–37]. Meanwhile, the majority of egg yolk proteins are synthesized from liver and transferred from the bloodstream when the yolk starts to mature in the ovary. Chicken liver expresses high levels of α-2,6 sialyltransferase (ST6GAL1), and it is assumed that chicken liver has very low levels of fucosyltransferase (FUT8); this is because most N-glycans in egg yolk protein are afucosylated[38,39]. Therefore, it is expected that proteins synthesized in chicken liver have a highly α-2,6 sialylated and afucosylated N-glycan structure, and that these proteins will circulate in the bloodstream and, ultimately, accumulate in egg yolk[40]. Therefore, if hIgG1 Fc can be synthesized in chicken liver specific manner, hIgG1 Fc can be obtained from serum and egg yolk with high levels of α-2,6 sialylation and afucosylation, and this will be beneficial to anti-inflammatory activity.

Here, we report the development of genome-edited chickens that secrete human IgG1 Fc in liver-specific manner. To do this, we tagged the human IgG1 Fc coding sequence to the *Albumin* (*ALB*) gene, which is a major serum protein and expressed in the liver, and generated genome-edited chickens (hereafter referred to as *ALB::hIgG1 Fc* chickens). For tagging, we used self-cleavage 2A peptide of thosea asigna virus (T2A) between ALB and IgG1 Fc coding sequences, thereby IgG1 Fc can be separated from ALB after translation. Using this strategy, we efficiently produced highly α-2,6 sialylated and afucosylated human IgG1 Fc in chicken serum and, ultimately, in egg yolk. The recombinant human IgG1 Fc (rhIgG1 Fc) derived from *ALB::hIgG1 Fc* chickens has improved FcγRIIIA blocking activity and successfully recapitulated anti-inflammatory activities of IVIG in vivo. Therefore, we show here that genome-edited chickens which express hIgG1 Fc in liver-specific manner can be a potential alternative source of IVIG to human plasma.

## Results

**Generation of *ALB::hIgG1 Fc* genome-edited chickens**. To accumulate rhIgG1 Fc in serum and egg yolk, we generated *Albumin (ALB)::hIgG1 Fc* genome-edited chickens using CRISPR/Cas9-NHEJ-mediated genome editing technology in White Leghorn (WL) primordial germ cells (PGCs) as previously reported[41,42]. To introduce the hIgG1 Fc coding sequence (CDS) into the *ALB* gene, we constructed a CRISPR/Cas9 plasmid that recognizes intron 13 of the *ALB* gene, and a donor plasmid containing intron 13 and exon 14 (without the stop codon) of the *ALB* gene, followed by the T2A sequence and the hIgG1 Fc CDS, which contains the hIgG1 hinge and the CH2 and CH3 domains (Fig. 1a). T2A is one of the viral 2A self-cleavage peptides that induce the missing of peptide bond formation during translation, thereby enabling simultaneous expression of two different proteins[43]. Although the cleavage efficiency of 2A peptides was not 100%[44], we intended to separate rhIgG1 Fc from ALB using T2A without affecting ALB formation and secretion, thereby preventing unknown harmful effect caused by ALB depletion.

After transfection of the CRISPR/Cas9 and donor plasmids, genome-edited PGCs were selected by puromycin selection. Donor plasmid integration into selected PGCs was validated by PCR and sequence analysis of the 5' and 3' junction between the endogenous and donor plasmid sequences; this confirmed that the donor plasmid was integrated successfully at the target site, with several indels (Fig. 1b, c). These established genome-edited WL PGCs were transplanted into Korean Ogye (KO) recipient embryos. After sexual maturation, two recipient KO males were

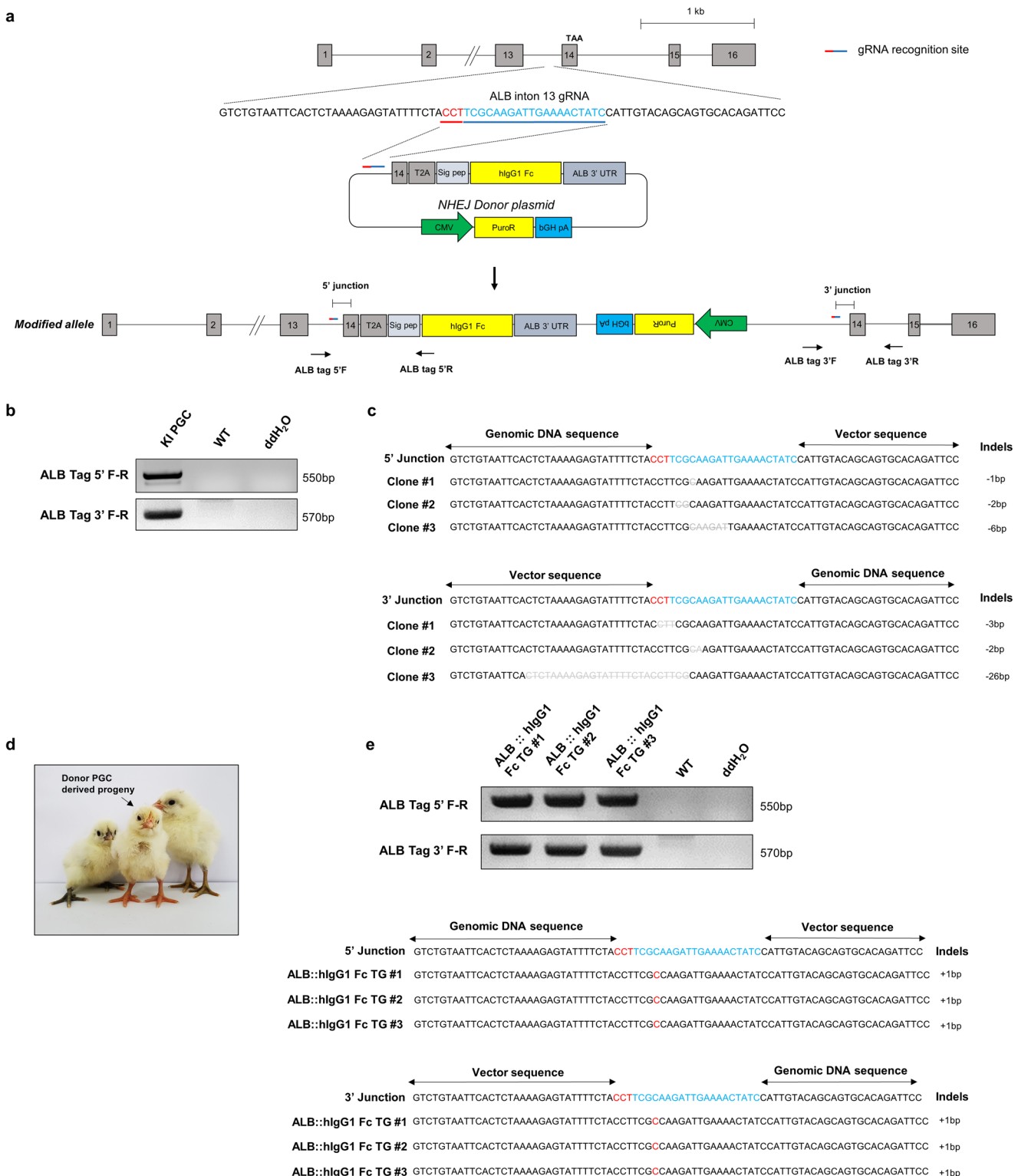

**Fig. 1 Generation of *ALB::hIgG1 Fc* genome-edited chickens. a** Schematic illustration of the donor plasmid containing the human IgG1 Fc (hIgG1 Fc) coding sequence for tagging to *Albumin (ALB)* gene. The donor vector includes *ALB* intron 13, *ALB* exon 14 without the stop codon, the T2A sequence, the *ALB* signal peptide (sig pep) coding sequence, the hIgG1 Fc coding sequence, the *ALB* 3′ UTR, and a puromycin resistance (PuroR) gene. When the donor vector and single guide RNA (sgRNA), which targets *ALB* intron 13, were co-transfected into primordial germ cells (PGCs), the donor vector was inserted into the sgRNA target site. The resulting modified allele contains *ALB* exon 14 without the stop codon, the T2A coding sequence, the *ALB* signal peptide coding sequence, and the hIgG1 Fc coding sequence. **b** Knock-in (KI) validation of PGCs after puromycin selection using primers specific for the 5′ junction and 3′ junction. **c** Sequence analysis of three TA-cloned PCR products of the 5′ junction and 3′ junction of genome-edited PGCs. **d** Production of donor PGC-derived progeny by testcross. **e** Sequence analysis of three TA-cloned PCR products of the 5′ junction and 3′ junction of three *ALB::hIgG1 Fc* genome-edited G1 progeny.

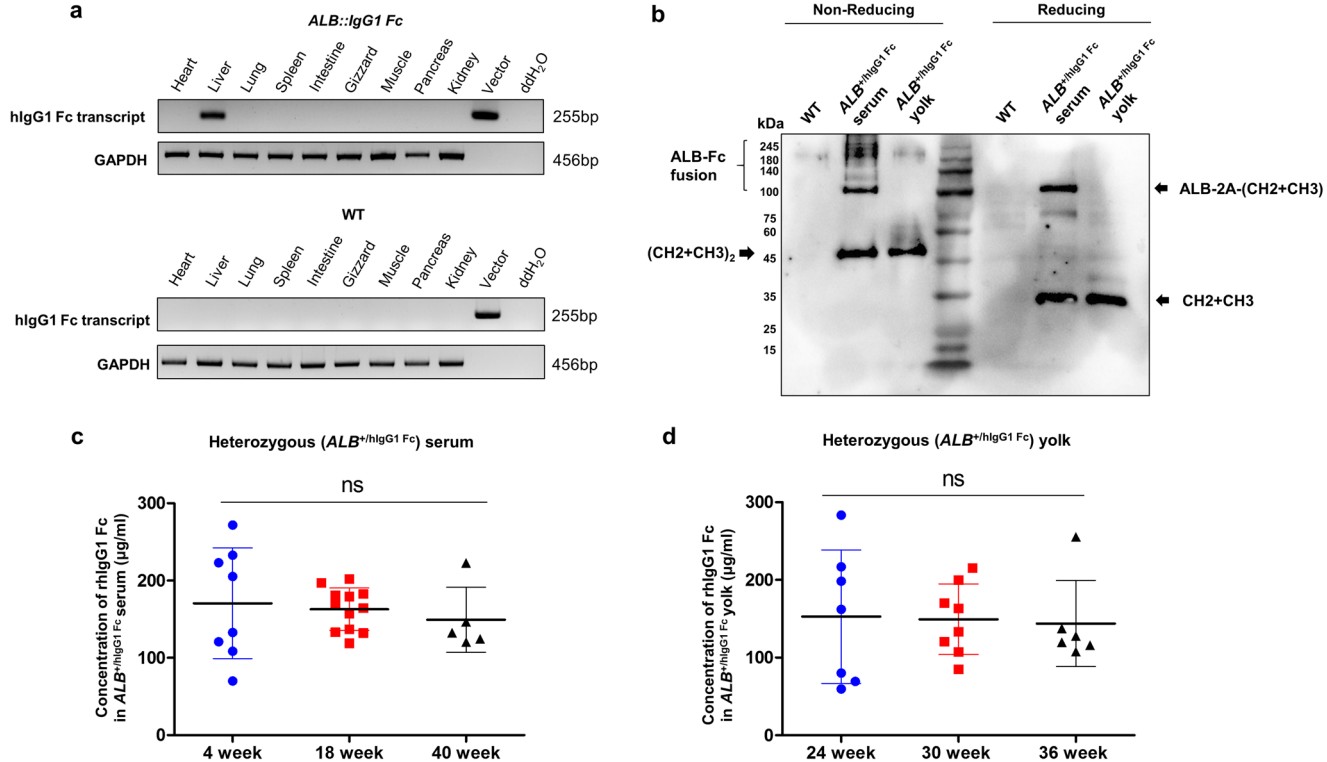

**Fig. 2 Accumulation of rhIgG1 Fc in serum and egg yolk. a** Validation of expression of the hIgG1 Fc transcript by RT-PCR of tissue from several organs obtained from wild-type and *ALB::hIgG1 Fc* genome-edited chickens. **b** Validation of rhIgG1 Fc secretion into serum and egg yolk of *ALB::hIgG1 Fc* genome-edited chickens by western blotting. β-mercaptoethanol were added to samples to break disulfide bonds (reducing conditions). Wild-type chicken serum was used as negative control. **c** Concentration of rhIgG1 Fc in serum from heterozygous $ALB^{+/hIgG1\ Fc}$ chicks at 4, 18, and 40 weeks after hatching. Each dot represents individual chickens. ($n = 5$–12 of independent samples) **d** Concentration of rhIgG1 Fc in heterozygous $ALB^{+/hIgG1\ Fc}$ egg yolks at 24, 30, and 36 week after hatching. Each dot represents individual egg yolks. ($n = 6$–8 of independent samples) Differences among the groups were determined by one-way ANOVA. ns, not significant. Error bars represent standard deviation.

testcrossed with wild-type WL hens and the donor PGC-derived G1 offspring were hatched (Fig. 1d and Supplementary Table 1). From the donor PGC-derived G1 offspring, we obtained *ALB::hIgG1 Fc* progeny in which integration of the donor plasmid was validated by 5' and 3' junction-specific PCR and sequence analysis of genomic DNA. In genome-edited progeny, the donor plasmid was integrated successfully, with a one base pair insertion mutation at both the 5' and 3' junction sites (Fig. 1e).

Next, one sexually mature heterozygous G1 rooster ($ALB^{+/hIgG1\ Fc}$) was mated with wild-type hens ($ALB^{+/+}$), and G2 *ALB::hIgG1 Fc* offspring ($ALB^{+/hIgG1\ Fc}$) were hatched. After sexual maturation, these heterozygous G2 progeny were mated and G3 progeny were hatched successfully. Genotyping of the G3 progeny confirmed that homozygous *ALB::hIgG1 Fc* progeny ($ALB^{hIgG1\ Fc/hIgG1\ Fc}$) were generated successfully according to Mendelian inheritance (Supplementary Fig. 1a, b). Although the concentration of ALB in serum showed tendency to decrease in homozygous chickens, we observed that homozygous chickens also secrete ALB into blood, are healthy, reach sexual maturity, and lay eggs (Supplementary Fig. 2a–c, e). These results demonstrate that *ALB::hIgG1 Fc* chickens can be produced successfully, and that they reach sexual maturity and lay eggs.

**Accumulation of rhIgG1 Fc in serum and egg yolk.** To identify the expression pattern of hIgG1 Fc transcripts in *ALB::hIgG1 Fc* chickens, we extracted RNA from several organs and confirmed the expression of hIgG1 Fc transcripts by RT-PCR. As a result, we identified strong expression of rhIgG1 Fc transcripts in the liver

(Fig. 2a), suggesting that rhIgG1 Fc was transcribed successfully in the liver-specific manner.

Next, we sampled serum and egg yolk and validated secretion of rhIgG1 Fc by western blotting (Fig. 2b). The data confirmed secretion of hIgG1 Fc into the bloodstream and egg yolk (Fig. 2b). Under non-reducing conditions, a band consistent with rhIgG1 Fc ($(CH2 + CH3)_2$) was detected in both serum and yolk. Under reducing conditions, a band of around 35 kDa was detected, which is bigger than the expected molecular mass of deglycosylated CH2 + CH3 (25 kDa) (Fig. 2b). This result shows that rhIgG1 Fc is secreted into the serum and egg yolk in its glycosylated form. Interestingly, in serum, a large band of around 110 kDa (consistent with the molecular mass of ALB + T2A + CH2 + CH3) was also detected under reducing conditions, as well as the ALB-Fc fusion (a large band of > 100 kDa) under non-reducing conditions, suggesting that T2A cleavage was not completely performed (Fig. 2b). In contrast to serum, only rhIgG1 Fc was detected egg yolk under both reducing and non-reducing conditions (Fig. 2b).

Next, we checked the rhIgG1 Fc concentration in serum and egg yolk of G2 heterozygous $ALB^{+/hIgG1\ Fc}$ hens. The average rhIgG1 Fc concentration in the serum of heterozygous $ALB^{+/hIgG1\ Fc}$ progeny at 4, 18, and 40 weeks after hatching was $170.70 \pm 71.65$ ($n = 8$) µg/ml, $162.98 \pm 27.34$ ($n = 12$) µg/ml, and $149.39 \pm 42.34$ µg/ml ($n = 5$), respectively (Fig. 2c). The rhIgG1 Fc concentration in egg yolk of G2 heterozygous $ALB^{+/hIgG1\ Fc}$ hens showed an average of $152.85 \pm 85.87$ ($n = 7$) µg/ml, $149 \pm 45.52$ ($n = 8$) µg/ml, and $143.89 \pm 55.57$ ($n = 6$) µg/ml at 24, 30, and 36 weeks, respectively (Fig. 2d). Also, we confirmed

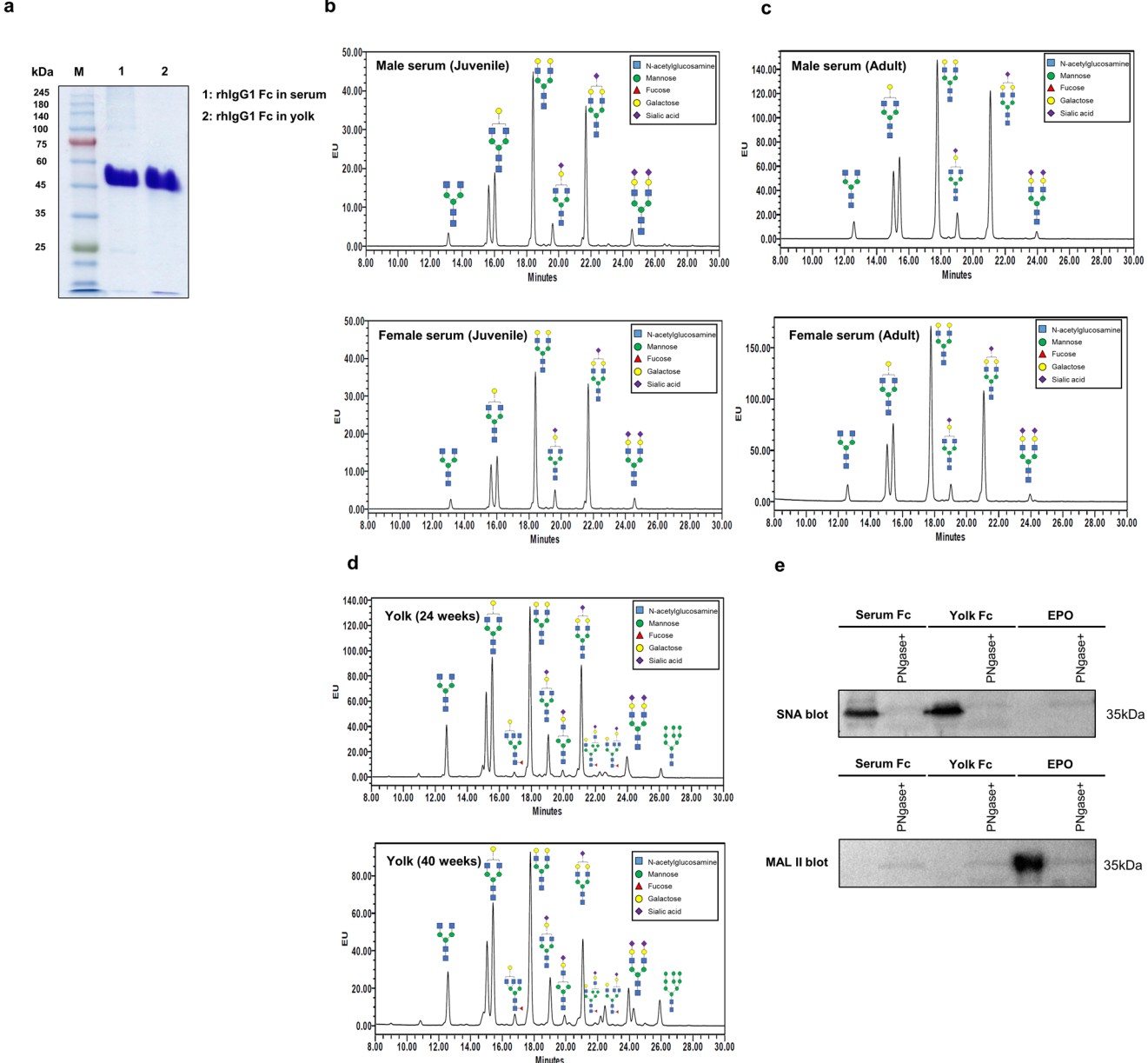

**Fig. 3 N-glycosylation pattern analysis of rhIgG1 Fc purified from *ALB:: hIgG1 Fc* genome-edited chickens. a** rhIgG1 Fc was purified from serum and yolk using a protein A column and size exclusion chromatography. Purified rhIgG1 Fc was run on a 10% SDS-PAGE gel, which was then stained with Coomassie brilliant blue solution. **b** The N-glycosylation profile of rhIgG1 Fc purified from juvenile serum, **c** adult serum, and **d** egg yolk at 24 and 40 weeks, as analyzed by UPLC-MS/MS. **e** *Sambucus nigra* (SNA) lectin and *Maackia amurensis* lectin II (MAL II) blot of rhIgG1 Fc purified from serum and egg yolk. CHO cell derived recombinant erythropoietin (EPO) which has only α-2,3 sialic acid was used as negative control for SNA blot and positive control for MAL II blot.

that rhIgG1 Fc is secreted into the bloodstream continuously, regardless of progeny in several generations (Supplementary Table 2). Furthermore, we observed that rhIgG1 Fc is also secreted and accumulated in the serum and yolk of homozygous $ALB^{hIgG1 \, Fc/hIgG1 \, Fc}$ hens (Supplementary Fig. 2d and e). These results show that *ALB::hIgG1 Fc* chickens efficiently secrete rhIgG1 Fc into serum and egg yolk.

**N-glycosylation pattern and plasma half-life of rhIgG1 Fc derived from *ALB::hIgG1 Fc* chickens**. To examine the pattern of N-glycosylation on rhIgG1 Fc derived from *ALB::hIgG1 Fc* chickens, we purified rhIgG1 Fc from serum and egg yolk using protein A affinity and size exclusion chromatography. rhIgG1 Fc was purified successfully from both serum and egg yolk. (Fig. 3a).

Next, N-glycans were removed by digestion with PNGase F and analyzed by UPLC-MS/MS. In the case of rhIgG1 Fc purified from serum, we identified six types of N-glycan, and all were bi-antennary and afucosylated complex forms (Fig. 3b–d). From this N-glycan profile, we determined the percentage of individual N-glycans (Tables 1 and 2). Serum from juvenile female and male chickens comprised about 2.5% degalactosylated G0 glycoforms, whereas the majority of N-glycans were terminally galactosylated and sialylated (Fig. 3b and Table 1). Among these, the total composition of N-glycans in juvenile female and male serum containing terminal sialic acid residues was 36.01% and 39.00%, respectively (Table 1). Similarly, in both adult female and male serum the degalactosylated G0 glycoform was present at around 3.3%, and terminal sialic acid was present at around 28.66% and

**Table 1 N-glycan pattern profiles of rhIgG1 Fc from juvenile (12 ~ 15 weeks) and adult (40 weeks) serum.**

| Glycan identification | | Percent to total intensity (%) | | | |
|---|---|---|---|---|---|
| | | Female serum (juvenile) | Male serum (juvenile) | Female serum (adult) | Male serum (adult) |
| G0 | | 2.59 | 2.39 | 3.30 | 3.20 |
| G1 | | 26.92 | 24.69 | 29.30 | 28.16 |
| A3G1F | | - | - | - | - |
| G2 | | 34.49 | 33.91 | 38.74 | 34.20 |
| G1S1 | | 4.61 | 4.73 | 3.15 | 4.91 |
| M3G1S1 | | - | - | - | - |
| G2S1 | | 27.86 | 31.47 | 24.26 | 28.07 |
| M5A1S1F | | - | - | - | - |
| A3G2S1F | | - | - | - | - |
| G2S2 | | 3.54 | 2.80 | 1.25 | 1.46 |
| M9 | | - | - | - | - |
| Terminal sialylation ratio (%) | | 36.01 | 39.00 | 28.66 | 34.44 |
| Fucosylation ratio (%) | | - | - | - | - |

■ N-acetylglucosamine
● Mannose
▲ Fucose
● Galactose
◆ Sialic acid

*A* N-acetylglucosamine, *F* Fucose, *G* Galactose, *M* Mannose, *S* Sialic acid.

34.44%, respectively (Fig. 3c and Table 1). These results show that *ALB::hIgG1 Fc* chickens secrete higher levels of sialylated and afucosylated rhIgG1 Fc into serum.

Next, we analyzed rhIgG1 Fc purified from egg yolk at 24 and 40 weeks. Initially, we expected that the N-glycan profile of rhIgG1 Fc in egg yolk would be the same as that in serum because egg yolk proteins are transported from the serum. Although the major N-glycan types were the same as in serum, we identified five additional minor N-glycan types: A3G1F, M3G1S1, M5A1S1F, A3G2S1F, and M9. These comprised 6.04% and 10.99% of total N-glycans in egg yolk at 24 and 40 weeks, respectively (Fig. 3d and Table 2). Among these, core fucosylated N-glycans were present as 3.53% and 5.71% of total N-glycans at 24 and 40 weeks, respectively (Table 2). The percentage of rhIgG1 Fc N-glycans containing terminal sialic acid residues was 32.07% and 30.77% at 24 and 40 weeks,

respectively (Table 2). These results show that the N-glycan profile of rhIgG1 Fc in egg yolk is largely the same as that in serum, but with some minor differences.

Next, to identify the major linkage type between terminal sialic acid and galactose residues, which is one of the major factors underlying the anti-inflammatory activity of hIgG1 Fc[11], we performed lectin blotting using HRP-labeled *Sambucus nigra* lectin (SNA) and *Maackia amurensis* lectin II (MAL II), which bind specifically to α-2,6- and α-2,3-linked sialic acid residues, respectively. SNA blots revealed a strong signal for both serum and yolk rhIgG1 Fc. However, the MAL II blot showed a negligible signal for both serum and yolk rhIgG1 Fc, suggesting that the major linkage type is α-2,6 (Fig. 3e). These results show that high levels of α-2,6 sialylated and afucosylated rhIgG1 Fc can be synthesized efficiently by *ALB::hIgG1 Fc* chickens and purified from both serum and egg yolk.

**Table 2 N-glycan pattern profiles of rhIgG1 Fc from egg (24 and 40 weeks).**

| Glycan identification | | Percent to total intensity (%) | |
|---|---|---|---|
| | | Yolk (24 weeks) | Yolk (40 weeks) |
| G0 | | 7.70 | 7.49 |
| G1 | | 28.00 | 31.4 |
| A3G1F | | 1.14 | 1.47 |
| G2 | | 26.26 | 24.95 |
| G1S1 | | 6.39 | 6.93 |
| M3G1S1 | | 1.12 | 1.36 |
| G2S1 | | 17.77 | 13.11 |
| M5A1S1F | | 1.04 | 1.40 |
| A3G2S1F | | 1.35 | 2.84 |
| G2S2 | | 4.40 | 5.13 |
| M9 | | 1.39 | 3.92 |
| Terminal sialylation ratio (%) | | 32.07 | 30.77 |
| Fucosylation ratio (%) | | 3.53 | 5.71 |

Legend:
- ▪ N-acetylglucosamine
- ● Mannose
- ▲ Fucose
- ● Galactose (yellow)
- ◆ Sialic acid

*A* N-acetylglucosamine, *F* Fucose, *G* Galactose, *M* Mannose, *S* Sialic acid.

Additionally, the serum half-life of *ALB::hIgG1 Fc* chicken-derived rhIgG1 Fc, HEK293T cell-derived recombinant Fc and IVIG was analyzed in C57BL/6 mouse. As a result, the half-life of *ALB::hIgG1 Fc* chicken-derived rhIgG1 Fc and HEK293T cell-derived recombinant Fc were determined as 39.14 and 36.37 h, respectively (Supplementary Fig. 3).

**Affinity of rhIgG1 Fc derived from *ALB::hIgG1 Fc* chickens for human Fc receptors**. Because various biological activities of IgG1 are due to the interaction between its Fc region and Fcγ receptors (FcγR), we measured the dissociation constant ($K_D$) for binding of rhIgG1 Fc to human type I Fcγ receptors (FcγRIA, FcγRIIA, and FcγRIIIA), dendritic cell-specific ICAM grabbing nonintegrin (DC-SIGN) (a receptor for α-2,6-sialylated Fc), and FcRn, by surface plasmon resonance (SPR). The $K_D$ values for binding of rhIgG1 Fc to FcRn, FcγRIA, and FcγRIIIA were $9.088 \times 10^{-9}$, $1.275 \times 10^{-9}$, and $9.980 \times 10^{-9}$ M, respectively (Fig. 4a). However, the $K_D$ values for binding of rhIgG1 Fc to FcγRIIA and DC-

SIGN could not be determined, mainly due to non-specific binding and low affinity (Supplementary Fig. 4 and Table 3). In addition, we measured the $K_D$ values for the binding of commercial IVIG to FcRn, FcγRIA, and FcγRIIIA: these were $2.142 \times 10^{-8}$, $2.480 \times 10^{-9}$, and $2.870 \times 10^{-7}$ M, respectively (Fig. 4a). From the evaluated $K_D$ values, we confirmed that the affinity of rhIgG1 Fc for FcγRIA and FcRn was 1.94 and 2.35-fold higher, respectively, than that of commercial IVIG (Table 3). In particular, the affinity of rhIgG1 Fc for FcγRIIIA was 28.75-fold higher than that of commercial IVIG (Table 3). These results suggest that rhIgG1 Fc has significantly higher monovalent affinity for FcγRIIIA than IVIG, most likely due to its high levels of afucosylation.

**rhIgG1 Fc derived from *ALB::hIgG1 Fc* chicken inhibits immune complex-mediated crosslinking of FcγRIIIA**. Because crosslinking of FcγRIIIA by immune complexes stimulates immune cells to secrete inflammatory cytokines and induce tissue

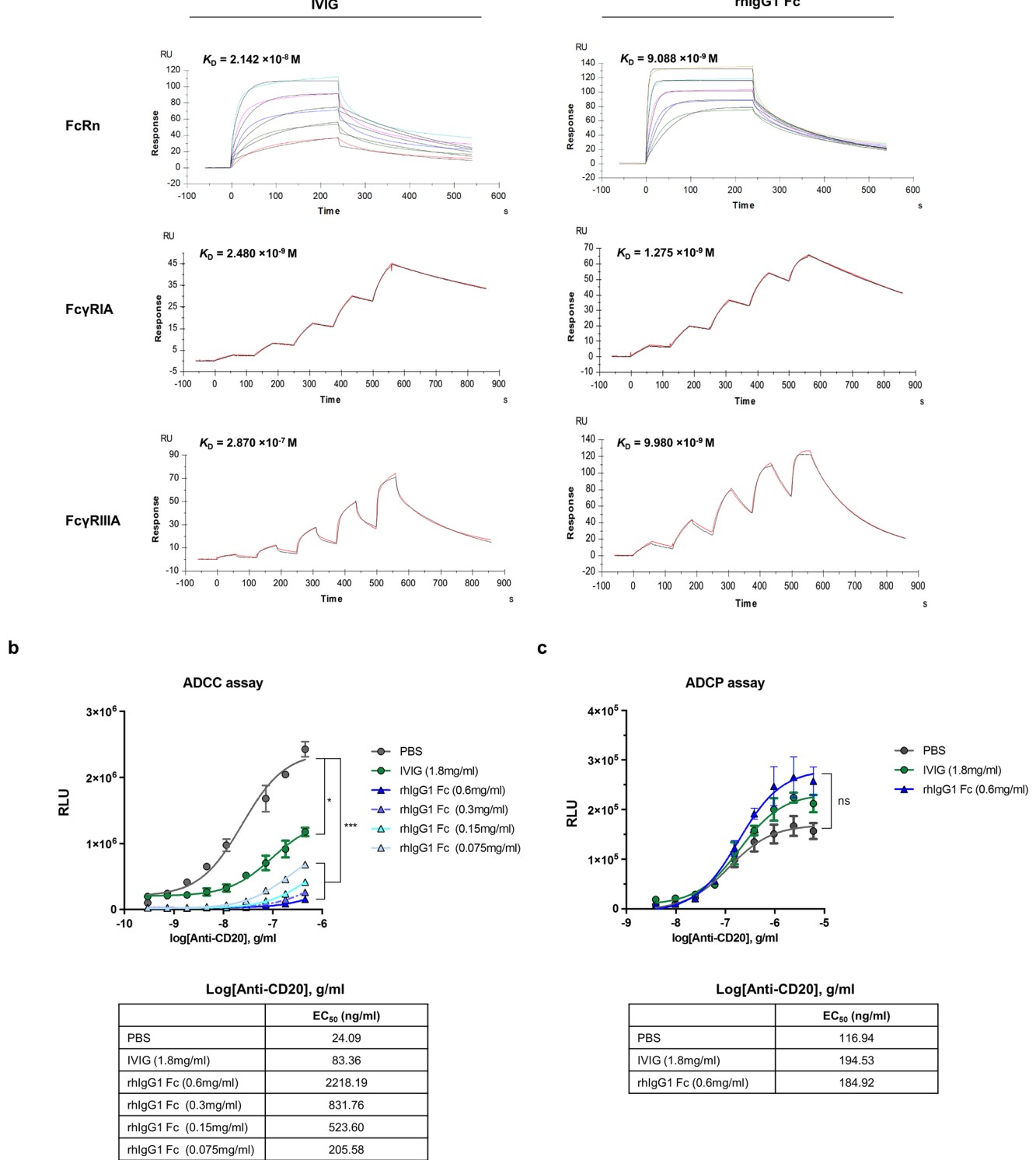

**Fig. 4 Affinity of rhIgG1 Fc purified from *ALB::hIgG1 Fc* chickens for human Fc gamma receptors and comparison with that of human IVIG.**
**a** Sensorgrams showing binding of IVIG and rhIgG1 Fc to FcRn, FcγRIA, and FcγRIIIA (Biacore analysis). Binding was evaluated using a steady state equilibrium model for FcRn, and single cycle kinetics for FcγRIA and FcγRIIIA. $K_D$ values are annotated on each sensorgram. **b** Antibody-dependent cellular cytotoxicity (ADCC) of an anti-CD20 antibody against WIL2-S human B-lymphoblasts in the presence of IVIG and rhIgG1 Fc. FcγRIIIA-expressing Jurkat cells were added as effector cells, and ADCC activity was evaluated by measuring luciferase activity. **c** Antibody-dependent cellular phagocytosis (ADCP) of an anti-CD20 antibody against Raji human lymphoma cells in the presence of IVIG and rhIgG1 Fc. FcγRIIA-expressing Jurkat cells used as effector cells, and ADCP activity was evaluated by measuring luciferase activity. ($n = 3$ of independent samples) Differences among groups were determined by one-way ANOVA. ns, not significant; * $P < 0.05$; and *** $P < 0.001$. Error bars represent standard deviation.

**Table 3 $K_D$ values between IVIG, rhIgG1 Fc and human FcγRs.**

| Analytes | FcγRI $K_D$ (M) | FcγRIIA $K_D$ (M) | FcγRIIIA $K_D$ (M) | FcRn $K_D$ (M) | DC-SIGN $K_D$ (M) |
|---|---|---|---|---|---|
| IVIG | $2.480 \times 10^{-9}$ | $9.644 \times 10^{-6}$ | $2.870 \times 10^{-7}$ | $2.142 \times 10^{-8}$ | N.D. |
| rhIgG1 Fc | $1.275 \times 10^{-9}$ | N.D. | $9.980 \times 10^{-9}$ | $9.088 \times 10^{-9}$ | N.D. |
| Fold change (IVIG/rhIgG1 Fc) | 1.94 | N.D. | 28.75 | 2.35 | N.D. |

*N.D.* not determined.

damage, FcγRIIIA blockade is a potent anti-inflammatory strategy; indeed, the anti-inflammatory activity of IVIG is mediated by blockade of FcγRIIIA via the Fc region[19,21,45–47]. Therefore, we asked whether rhIgG1 Fc inhibits immune complex-mediated crosslinking of FcγRIIIA and subsequent immune cell activation. To do this, we examined the human FcγRIIIA-mediated ADCC activity of an anti-CD20 antibody in the presence of IVIG or rhIgG1 Fc. WIL2-S lymphoblast cells, which express human CD20, and transgenic Jurkat cells, which express FcγRIIIA and luciferase reporter gene driven by nuclear factor of activated T-cell (NFAT)-response element, were co-incubated with a serially diluted anti-CD20 antibody in the presence of IVIG or rhIgG1 Fc. Binding of the immune complexed anti-CD20 antibody to FcγRIIIA expressed by Jurkat cells will induce crosslinking of FcγRIIIA and activate NFAT pathway, resulting in luciferase expression.

When IVIG was added at 1.8 mg/ml, the $EC_{50}$ value of the anti-CD20 antibody was 83.36 ng/ml, which was 3.46-fold higher than that measured in the PBS treatment group (Fig. 4b). This indicated that IVIG has FcγRIIIA blocking activity and hinders FcγRIIIA crosslinking mediated by immune complexes. Interestingly, when rhIgG1 Fc was added at 600 μg/ml, the same molar ratio as 1.8 mg/ml IVIG, the $EC_{50}$ value of the anti-CD20 antibody was 2218.19 ng/ml, which was 26.60-fold higher than that of the IVIG-treated group (Fig. 4b). This difference in fold-change is consistent with the difference in the affinity of IVIG and rhIgG1 Fc for FcγRIIIA (Fig. 4a and Table 3). When the concentration of rhIgG1 Fc decreased, the $EC_{50}$ value increased proportionately (Fig. 4b). These results indicate that rhIgG1 Fc is superior to commercial IVIG with respect to occupation of FcγRIIIA, and that it efficiently blocks binding of immune complexes to this receptor.

Also, we tried to confirm that rhIgG1 Fc blocks FcγRIIA, a major inducer of ADCP, using transgenic Jurkat cells that express FcγRIIA and luciferase reporter gene driven by NFAT-response element. The CD20 expressing Raji cells and transgenic Jurkat cells were co-incubated with a serially diluted anti-CD20 antibody in the presence of 1.8 mg/ml IVIG or 0.6 mg/ml rhIgG1 Fc. The ADCP induction levels were examined by measuring the luciferase activity in each group. However, we did not detect inhibitory effects on FcγRIIA crosslinking mediated by immune complexed anti-CD20 antibody in both 1.8 mg/ml IVIG and 0.6 mg/ml rhIgG1 Fc treatment group (Fig. 4c). This result is consistent with the relatively low affinity of IVIG and rhIgG1 Fc for FcγRIIA than FcγRIIIA (Table 3 and Supplementary Fig. 4).

**rhIgG1 Fc derived from *ALB::hIgG1 Fc* chickens has anti-inflammatory activity in vivo.** To verify whether rhIgG1 Fc blocks autoantibody-mediated platelet depletion, we used passive ITP mouse model in which platelets are depleted by injection of rat monoclonal antibody (MWReg30) recognizing murine platelet specific integrin $\alpha_{IIb}\beta_3$[48]. We administered IVIG or rhIgG1 Fc into mouse 2 h before injection of anti-platelet antibody MWReg30 and then analyzed platelet count after 4 h of MWReg30 injection (Fig. 5a). When IVIG was administrated at 1 g/kg and 0.33 g/kg, it prevented platelet depletion mediated by

MWReg30 (Fig. 5b). Similarly, when rhIgG1 Fc was administrated at 0.33 g/kg and 0.12 g/kg, which is the same molar ratio as 1 g/kg and 0.33 g/kg IVIG, it also prevented platelet depletion (Fig. 5c). However, when IVIG was administered at 0.1 g/kg, it did not prevent platelet depletion by MWReg30 (Fig. 5b). In contrast to IVIG, when rhIgG1 Fc was administered at 0.033 g/kg, which is the same molar ratio as 0.1 g/kg IVIG, it prevented platelet depletion by MWReg30 (Fig. 5c). These results indicate that rhIgG1 Fc efficiently blocks platelet depletion mediated by MWReg30 immune complex.

Because α-2,6 sialylated IgG1 Fc in IVIG exerts anti-inflammatory activity by expanding the regulatory T cell (Treg) population and promoting expression of FcγRIIB via $T_H2$ cytokines IL-33 and IL-4[12,13], we examined whether rhIgG1 Fc also recapitulate anti-inflammatory activity of α-2,6 sialylated IgG1 Fc by examining Treg population and expression of *IL-33*, *IL-4*, and *FcγRIIB* in the spleen. The results confirmed that the CD4[+]/ Foxp3[+] regulatory T cell population have a tendency to increase after 6 h treatment in both the IVIG and rhIgG1 Fc groups, although it does not reach statistically significance (Fig. 5d, e). Furthermore, we confirmed that expression of *IL-33*, *IL-4*, and *FcγRIIB* significantly increased in the IVIG and rhIgG1 Fc treatment groups (Fig. 5f). These results show that chicken-derived rhIgG1 Fc promotes expression of $T_H2$ cytokines and FcγRIIB, and thereby successfully recapitulating the mechanisms of anti-inflammatory activity mediated by α-2,6 sialylated IgG1 Fc in IVIG.

## Discussion
Here, we developed genome-edited chickens that produce rhIgG1 Fc in serum and egg yolk. We show that *ALB::hIgG1 Fc* chickens secrete rhIgG1 Fc into serum and egg yolk, and that rhIgG1 Fc derived from *ALB::hIgG1 Fc* chickens is a functional anti-inflammatory agent. Also, we suggested that genome-edited chickens can be used as an efficient platform for production of an IVIG alternative with enhanced anti-inflammatory activity. Furthermore, we suggested that targeting major serum protein coding genes expressed predominantly in chicken liver is an efficient approach to accumulating highly galactosylated, α-2,6 sialylated and afucosylated, biopharmaceuticals in serum and egg yolk.

We found that *ALB::hIgG1 Fc* chickens showed dominant expression of hIgG1 Fc transcripts in the liver, and that the N-glycosylation pattern harbored by hIgG1 Fc derived from *ALB::hIgG1 Fc* chickens showed high levels of α-2,6 sialylation and low levels of fucosylation. In particular, sialylated N-glycans comprised around 30% of total N-glycans. This is quite efficient sialylation ratio considering that the hIgG1 Fc is barely sialylated protein because of its structure[49]. In general, human IgG1 produced from mammalian expression system has barely sialylated Fc, and natural human IgG1 from human plasma has a low sialylation ratio (around 10% of total N-glycans)[26,27]. Also, human IgG produced from mammalian expression systems, as well as natural human IgG1 from plasma, contain high levels of core fucosylated Fc[26,27]. Therefore, *ALB::hIgG1 Fc* chickens are a unique system for producing hIgG1 Fc with abundant terminal sialylations and low levels of fucosylation. Additionally, we

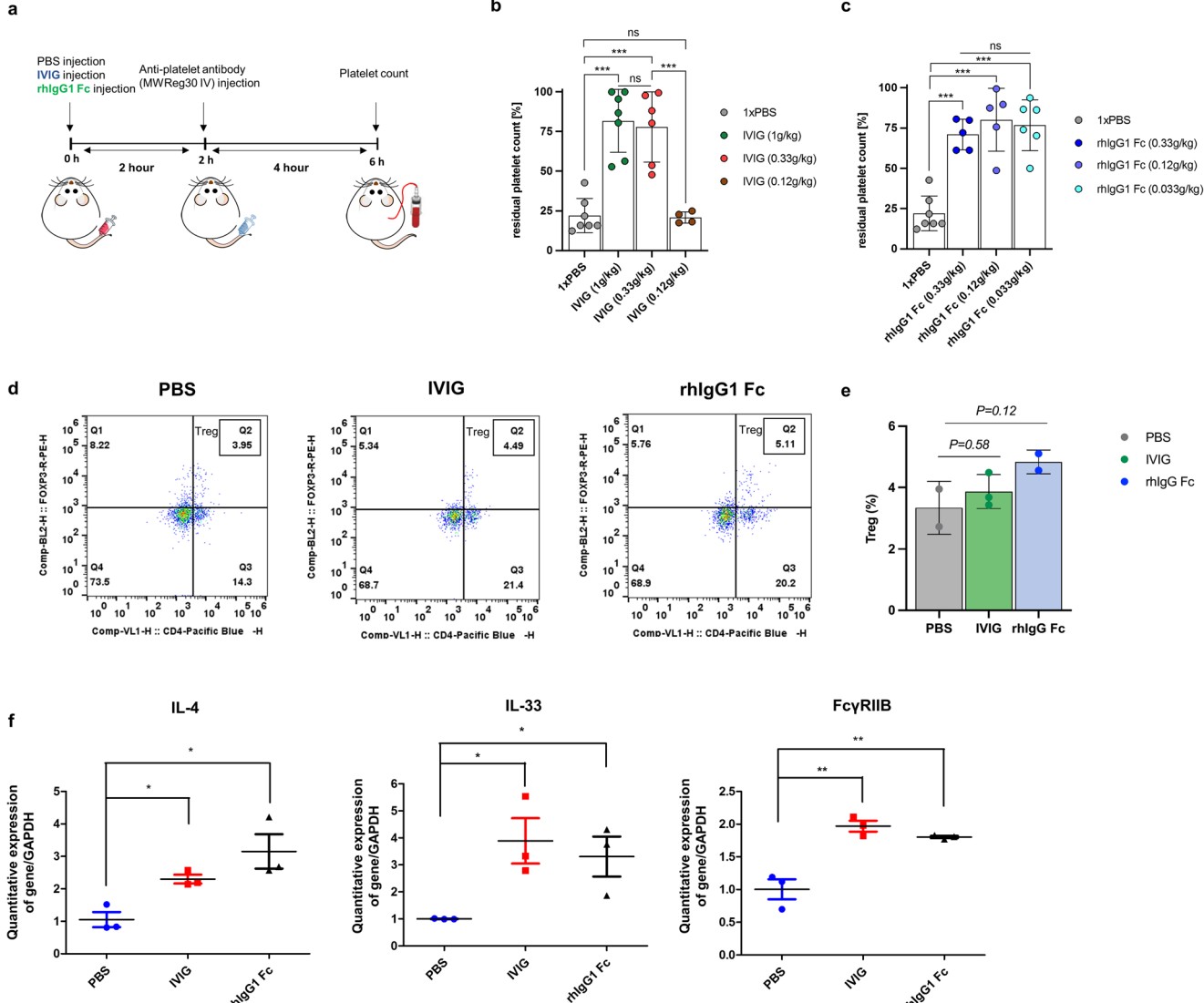

**Fig. 5 Use of a passive immunothrombocytopenia (ITP) mouse model to validate the anti-inflammatory activity of rhIgG1 Fc purified from *ALB::hIgG1 Fc* chickens. a** Schematic illustration of the in vivo experiment using a passive ITP mouse model. Two hours before injection of an anti-platelet antibody MWReg30 (0.1 μg/g), mice received an intraperitoneal injection of IVIG or rhIgG1 Fc. At 4 h post-injection of MWReg30, blood was extracted from the tail vein and platelets were stained by Rees and Ecker diluting fluid and counted under a phase-contrast microscope. **b** The residual platelet count was quantified at 4 h post-injection of MWReg30, which itself was injected 2 h after IVIG (1 g/kg, 0.33 g/kg, or 0.12 g/kg). Each dot represent individual mice. ($n = 4$–7 of independent samples). **c** The residual platelet count was quantified 4 h after injection of MWReg30, which was administered 2 h after hIgG1 Fc (0.33 g/kg, 0.12 g/kg, or 0.033 g/kg). ($n = 5$-7 of independent samples). **d** Representative flow cytometry plots demonstrating expansion of the endogenous CD4$^+$/Foxp3$^+$ Treg population in the spleen. Spleens from the PBS, IVIG (1 g/kg), rhIgG1 Fc (0.33 g/kg) treatment groups were analyzed. **e** Treg population percentages of spleens from the PBS, IVIG (1 g/kg), rhIgG1 Fc (0.33 g/kg) treatment groups after 6 h of treatment analyzed by flow cytometry. Each dot represents individual mice. **f** Total RNA was isolated from the spleen and used for quantitative real-time PCR to measure *IL-4, IL-33*, and *FcγRIIB* mRNA levels. Spleens from the PBS, IVIG (1 g/kg), rhIgG1 Fc (0.33 g/kg) treatment groups were analyzed. ($n = 3$ of independent samples) Differences among groups were determined by one-way ANOVA and a paired *t*-test. * $P < 0.05$, ** $P < 0.01$, and ***$P < 0.001$. Error bars represent standard deviation.

confirmed that the terminal sialic acid residues of rhIgG1 Fc derived from *ALB::hIgG1 Fc* chickens are mainly α-2,6 linked, which is critical for induction of anti-inflammatory activity[11]. Therefore, a chicken bioreactor is an efficient platform for producing hIgG1 Fc harboring beneficial N-glycans for inducing anti-inflammatory activity.

Our N-glycosylation pattern analysis detected minor N-glycans, which were originally unidentified in serum, in rhIgG1 Fc derived from egg yolk. This result is consistent with that of a previous report showing that several N-glycans, which were unidentified in serum IgY, were detected in yolk IgY; thus it

seems that minor modifications to N-glycans are performed when proteins are transported from serum to egg yolk[50]. Also, we did not detect ALB-Fc fusion protein in egg yolk (although it was originally detected in serum), suggesting that unprocessed ALB-Fc fusion protein was not transferred to egg yolk, or that cleavage of Fc from the fusion partner occurred during the transportation process, as previously reported[33].

When we analyzed the affinity of rhIgG1 Fc for human FcγRs, we found that rhIgG1 Fc has significantly higher affinity for FcγRIIIA due to low levels of fucosylation. Therefore, rhIgG1 Fc inhibited FcγRIIIA crosslinking mediated by immune complexes

more potently than commercial IVIG. These results indicate that rhIgG1 Fc efficiently blocks FcγRIIIA in a monovalent manner. This is important because multivalent blocking of FcγRIIIA by monoclonal antibodies induces adverse side effects mediated by FcγRIIIA crosslinking, albeit while potently recapitulating the anti-inflammatory activity of IVIG and showed efficacy in refractory ITP[22,51]. A recent study reported that monovalent blocking of FcγRIIIA by a Fab fragment circumvents toxicity mediated by FcγRIIIA crosslinking[47]. In line with this, the low-fucosylated hIgG1 Fc derived from *ALB::hIgG1 Fc* chickens monovalently blocked FcγRIIIA with higher affinity than human IVIG, and induced anti-inflammatory response without associated toxicity. Therefore, the chicken bioreactor system may be an optimal production platform for hIgG1 Fc with enhanced FcγRIIIA blocking and subsequent anti-inflammatory activity due to low fucosylation.

We also expect that rhIgG1 Fc will have higher affinity for DC-SIGN because the latter was suggested as receptor to recognize α-2,6 sialylated Fc[52]. Unfortunately, in our SPR analysis setting, the affinity of IVIG and rhIgG1 Fc to human DC-SIGN could not be measured. This results are in accordance with recent report showing that human DC-SIGN showed negligible binding affinity to α-2,6 sialylated Fc in flow cytometry and SPR analysis[53]. Other methods such as cell-based ELISAs may be required to measure the affinity between sialylated Fc and DC-SIGN.

To confirm that rhIgG1 Fc derived from genome-edited chickens also has anti-inflammatory activity in vivo, we administrated rhIgG1 Fc to a passive ITP mouse model. We found that rhIgG1 Fc derived from genome-edited chickens exerts anti-inflammatory activity in these mice, even at lower doses. Because previously reports showed that enhancement of α-2,6 sialylation in Fc region significantly enhance anti-inflammatory activity of IVIG, enhanced anti-inflammatory activity of rhIgG1 Fc derived from genome-edited chickens might be due to increased α-2,6 sialylation of Fc than plasma-derived IVIG[10–12,54]. Additionally, low-fucosylated rhIgG1 Fc also may affect to improved anti-inflammatory activity of rhIgG1 Fc and this result is in accordance with recent study showing that nonfucosylated and galactosylated IgG have improved anti-inflammatory activity in vivo[55].

In our study, both IVIG and rhIgG1 Fc have a tendency to expand the Treg population and promoted expression of IL-33, IL-4, and FcγRIIB, suggesting that hIgG1 Fc derived from *ALB::hIgG1 Fc* chickens successfully recapitulates the anti-inflammatory activity of α-2,6 sialylated IgG1 Fc in IVIG. Based on these results, we suggested here that the *ALB::hIgG1 Fc* chickens have the potential to be one of the supply source of IVIG alternative and will solve limitations of IVIG therapy related to supply shortages. Furthermore, we proposed that the N-glycosylation characteristics of recombinant proteins derived from *ALB::hIgG1 Fc* chickens are optimal for production of human liver derived blood products such as blood clotting factors, because human blood factors also have an N-glycosylation profile comprising highly α-2,6 sialylated and low-fucosylated forms, which is identical to that of chicken liver derived proteins[56]. Finally, chickens do not produce non-human glycans; thus, genome-edited chickens can be an optimal platform for production of humanized recombinant human blood products. Otherwise, highly galactosylated, sialylated, and low-fucosylated glycoforms produced in egg yolk may result in monoclonal antibodies with not only enhanced Fc effector functions (such as CDC and ADCC activity), but also increased serum half-life[57].

Conclusively, we demonstrated here that chicken bioreactor system which targeted major serum protein gene efficiently accumulates hIgG1 Fc in serum and egg yolk and this approach has a potential to be widely applied to produce various biopharmaceuticals with enhanced efficacy via optimal N-glycosylation pattern.

## Methods

**Experimental animals and animal care.** The management and experimental use of chickens and mouse were approved by the Institutional Animal Care and Use Committee (IACUC), Seoul National University (SNU-190401-1-2 and SNU-210726-1-1). The experimenetal animals were cared according to a standard management program at the University Animal Farm and Institute of Laboratory Animal Resources, Seoul National University.

**Construction of CRISPR/Cas9 expression plasmids and donor plasmids.** The CRISPR/Cas9 vector targeting intron 13 of chicken *ALB* gene was constructed using the PX459 vector (Addgene plasmid #62988). To insert guide RNA (gRNA) sequences into the CRISPR/Cas9 plasmid, sense and antisense oligonucleotides were designed and synthesized (Bioneer, Daejeon, Korea). These oligonucleotides were annealed under the following thermocycling conditions: 30 sec at 95 °C, 2 min at 72 °C, 2 min at 37 °C, and 2 min at 25 °C. For targeted tagging of the hIgG1 Fc to the 3' end of the *ALB* gene, the donor plasmid containing intron 13 and exon 14 of *ALB* gene and T2A, followed by the hIgG1 Fc coding sequence and bovine growth hormone polyadenylation site was synthesized in the pBHA vector backbone (Bioneer, Daejeon, Korea). The oligonucleotides used for construction of CRISPR/Cas9 vector were listed in Supplementary Table 3.

**Establishment of *ALB::hIgG1 Fc* genome edited PGCs.** For PGC line establishment, the White Leghorn (WL) male PGCs were maintained and sub-passaged on knockout DMEM (Invitrogen, Carlsbad, CA) supplemented with 20% FBS (Invitrogen), 2% chicken serum (Sigma-Aldrich, St.Louis, MO), 1× nucleosides (Millipore, Temecula, CA), 2 mM l-glutamine, 1× nonessential amino acids, β-mercaptoethanol, 10 mM sodium pyruvate, 1× antibiotic–antimycotic (Invitrogen), and human basic fibroblast growth factor (10 ng/ml; Sigma-Aldrich). Chicken PGCs were cultured in an incubator at 37 °C under an atmosphere of 5% CO2 and 60–70% relative humidity. The PGCs were sub-cultured onto mitomycin-inactivated mouse embryonic fibroblasts at 5- to 6-day intervals via gentle pipetting. For genome editing in chicken PGCs, CRISPR/Cas9 plasmids (3 μg) and donor plasmids (3 μg) were co-introduced into $1 \times 10^5$ cultured PGCs with 6 μl of Lipofectamine 2000 reagent suspended in 1 ml OptiMEM. Then, 4 h after transfection, the transfection mixture was replaced with PGC culture medium. Puromycin (1 μg/ml) was added to the culture medium 1 days after transfection. After 2 days of puromycin treatment, selected PGCs were washed and replaced fresh PGC culture medium and expanded for 4 to 6 weeks. Targeted insertion of donor plasmid were confirmed by genomic DNA PCR analysis. The primers used in this study were listed in Supplementary Table 3.

**Production of *ALB::hIgG1 Fc* genome-edited chicken.** To produce genome-modified chickens, a window was cut at the sharp end of the Korean Ogye (KO) recipient egg, and more than 3000 genome-modified WL PGCs were transplanted into the dorsal aorta of Hamburger and Hamilton stages 14–17 recipient embryos. The egg window was sealed with paraffin film, and the eggs were incubated with the pointed end down until hatching. The hatched chicks were raised in the institutional animal farm. After sexual maturation, the recipient roosters were mated with WT female chickens. Genome-modified progeny were identified based on offspring feather color and subsequent genomic DNA analysis and sequencing.

**Quantification of hIgG1 Fc in serum and egg yolk by ELISA.** To quantify rhIgG1 Fc in serum, whole blood of $ALB^{+/hIgG1 Fc}$ and $ALB^{hIgG1 Fc/hIgG1 Fc}$ chicken were collected and 0.5 M ethylenediaminetetraacetic acid (EDTA) solution were added to collected blood to prevent blood coagulation. Collected blood samples were centrifuged at 2000 rpm, 10 min, 4 °C and supernatants were collected and used for ELISA analysis. To quantify rhIgG1 Fc in egg yolk, eggs laid from four G2 $ALB^{+/hIgG1 Fc}$ chicken was collected and egg yolks were separated from egg whites. The collected serum and yolk samples were diluted in DW and quantified by ELISA using a human IgG ELISA kit (ab100547, Abcam, Cambridge, UK), according to manufacturer's instruction. The measured concentrations were divided as three to adjust standard IgG molecular weight to rhIgG1 Fc.

**Purification of Human IgG1 Fc from serum and egg yolk.** For human IgG1 Fc purification from transgenic chicken serum, 4 M ammonium sulfate was added slowly to chicken serum. The mixture was stirred for overnight at 4 °C, and then centrifuged for 30 min at 4 °C at 10,000 g. The pellet was resuspended in equal volume of 1×PBS to the original serum volume. Then, the mixture was dialyzed against 20 mM sodium phosphate buffer (pH 7.2). The sample was loaded onto a Protein A column (GE Healthcare Bio-Sciences, Uppsala, Sweden), and the protein was eluted with 100 ml gradient of 100 mM Citric acid (pH 2.8). The protein was further purified and fractionated by size-exclusion chromatography (SEC) using a HiLoad Superdex 75 Column (GE Healthcare Bio-Sciences) pre-equilibrated with 20 mM Tris-HCl, 175 mM NaCl (pH 7.4). For human IgG1 Fc purification from egg yolk, egg yolk was separated and diluted with 9 volumes of distilled water. The

sample was freezed at -20 °C overnight and thawed at room temperature. After thawing, the supernatant fluid was collected and filtered. Then, the sample was loaded onto a Protein A column (GE Healthcare Bio-Sciences), and the protein was eluted with 100 ml gradient of 100 mM Citric acid (pH 2.8).

**Western blotting.** For western blot of rhIgG1 Fc in serum and yolk, the $ALB^{+/}$ $^{hIgG1\ Fc}$ chicken serum was diluted in distilled water (DW) and the $ALB^{+/hIgG1\ Fc}$ chicken egg yolk was pre-treated by freezing and thaw method as described in *Purification of Human IgG1 Fc from serum and egg yolk* section. The wild type chicken serum was used as control. The prepared sample was electrophoresed on 10% polyacrylamide gel, transferred to a polyvinylidene fluoride membrane (Millipore, Billerica, MA). After transfer, polyvinylidene fluoride membrane was blocked with 3% skim milk for 1 h at room temperature (BD Biosciences, San Jose, CA). The blocked membrane was incubated with goat-anti human IgG primary antibody (10319, Alpha diagnostic, San Antonio, TX) at 4 °C overnight. The primary antibody were washed with 0.1% PBST buffer three times and the membrane were incubated with horseradish peroxidase (HRP)-conjugated donkey anti-goat IgG secondary antibody (sc-2020, Santa Cruz Biotechnology, Dallas, TX, USA) for 1 h in room temperature. After washing secondary antibody, HRP enzyme activity was detected by adding ECL western blotting substrate (GE Healthcare Bio-Sciences).

**N-glycan pattern analysis by UPLC-MS/MS.** N-glycan analysis of rhIgG1 Fc derived from *ALB::hIgG1 Fc* chicken was performed by UPLC/MS-MS. Briefly, purified rhIgG1 Fc was incubated with PNGase F for 16 h at 37 °C. Deglycosyated rhIgG1 Fc was precipitated using ethanol and centrifuged at 10,000 g for 10 min. The supernatant containing released N-glycan was transferred to a new tube and dried completely using Speed-Vac concentrator. The dried sample was labelled with procainamide for fluorescence analysis. The labeled N-glycan sample was analyzed and quantified using UPLC/MS-MS. An ACQUITY UPLC BEH Glycan column (1.2 × 150 mm, 1.7 µm; Waters, New Castle, DE) with a fluorescence detector (Waters iClass UPLC) was used for the separation and detection of N-glycans. The LC conditions were as follows: flow rate (0.5 mL/min), column temperature (60 °C), mobile phase buffer A (100 mM ammonium formate, pH 4.5), buffer B (100% acetonitrile), injection volume (8 mL), linear gradient (75–60% B for 46.5 min, 60–0% B for 1.5 min, 0% B for 1 min, 0–75% B for 1 min, and 75% B for 13 min). A high-resolution mass spectrometry, triple-TOF MS (AB SCIEX, Concord, Ontario, Canada), was used for N-glycan identification. The N-glycan distribution was analyzed with Empower (Waters).

**Lectin blotting.** The purified rhIgG1 Fc and CHO cell-derived recombinant EPO (Cat No. 100-64, Peprotech) were incubated in Glycoprotein Denaturing Buffer (New England Biolabs, Ipswich, MA, USA) at 100 °C for 10 min. Then, the sample were cleaved by adding PNGase F (New England Biolabs) according to the manufacture's protocols. The purified and deglycosylated rhIgG1 Fc was electrophoresed on 10% polyacrylamide gel and transferred to a polyvinylidene fluoride membrane (Millipore) for lectin blotting. The membrane was blocked with 3% bovine serum albumin (Sigma-Aldrich) in Tris-buffered saline (20 mM Tris, 0.5 M NaCl, pH 7.5; TBS) at 4 °C overnight. Subsequently, the membrane was incubated with 1.0 µg/ml of biotinated lectins SNA/EBL (α2,6-SA specific; Vector Laboratories, Burlingame, CA, USA) and MAL II (α2,3 SA specific; Vector Laboratories) in TBST (TBS containing 0.05% Tween 20) for 1 h. After washing twice with TBST buffer, the membranes were incubated with horseradish peroxidase-conjugated avidin (VECTASTAIN ABC kit; Vector Laboratories) for 1 h. The membrane was washed with TBST buffer, and staining was performed with ECL Western blot detection reagents (GE Healthcare Bio-Sciences).

**SPR analysis.** For SPR analysis, IVIG was purchased from GC pharma, Yong-in, Korea. Human FcγRs were purchased from Sino Biological Inc, Beijing, China (FcγRIA-10256-H08H; FcγRIIA 167 Arg-10374-H08H; FcγRIIIA F176V-10389-H08H1; FcRn-CT009-H08H; DC-SIGN-10200-H01H). rhIgG1 Fc was purified from juvenile *ALB::hIgG1 Fc* chicken serum. SPR analysis were performed using Biacore T200 (GE Healthcare, Chicago, IL, USA). FcγRs were immobilized on CM5 sensor chip (BR-1005-30, GE Healthcare). IVIG and rhIgG1 Fc were injected into the FcγRs coated sensor chip. The kinetic assays were performed by three-fold serial dilution for FcγRIA, FcγRIIA, FcγRIIIA, DC-SIGN and two-fold serial dilution for FcRn. For FcγRIA and DC-SIGN, dilution started from 300 nM. For FcγRIIA, dilution started from 20 µM. For FcγRIIIA, dilution started from 3 µM of IVIG and 300 nM of hIgG1 Fc. For FcRn, dilution started from 500 nM of IVIG and 250 nM of rhIgG1 Fc. Regeneration was conducted by injection of 10 mM Glycine for FcγRIA, FcγRIIA, FcγRIIIA, DC-SIGN and 100 mM Tris-HCl for FcRn. The dissociation constant ($K_D$) for monomeric binding of IVIG and rhIgG1 Fc to FcγRs were calculated by single cycle kinetics model for FcγRIA, FcγRIIA, FcγRIIIA and multicycle kinetics model for FcRn, DC-SIGN.

**ADCC and ADCP assays.** ADCC assays were performed using an ADCC Reporter Bioassay Complete Kit (G7014, Promega, Madison, WI, USA) according to manufacturer's protocol. Briefly, CD20-positive WIL2-S cells were resuspended in RPMI 1640 and low IgG human serum and seeded at 10,000 cells per well in an opaque 96-well culture plate containing genetically engineered FcγRIIIA expressing effector Jurkat T cells. These genetically engineered Jurkat T cells also have luciferase reporter gene which expression is driven by NFAT response element. Then, IVIG (GC pharma) or rhIgG Fc and serially diluted anti-CD20 antibody were added and incubated for 6 h at 37 °C, 5% $CO_2$. Crosslinking of FcγRIIIA and activation of NFAT signaling was determined using the Bio-Glo Luciferase assay, and measurements were taken on a microplate reader.

ADCP assays were performed using and ADCP Reporter Bioassay Complete Kit (G9901, Promega) according to manufacturer's protocol. Briefly, CD20-positive Raji cells were resuspended in RPMI 1640 and low IgG human serum and seeded at 10,000 cells per well in an opaque 96-well culture plate containing genetically engineered FcγRIIA expressing effector Jurkat T cells. These genetically engineered Jurkat T cells also have luciferase reporter gene which expression is driven by NFAT response element. Then, IVIG or rhIgG Fc and serially diluted anti-CD20 antibody were added and incubated for 6 h at 37 °C, 5% $CO_2$. Crosslinking of FcγRIIA and activation of NFAT signaling was determined using the Bio-Glo Luciferase assay, and measurements were taken on a microplate reader.

**In vivo experiments.** 8 weeks of female C57BL/6 mouse were used in vivo experiment. Before start the experiment, blood samples of each mice were collected from tail vein. Platelets were stained with 1% brilliant cresyl blue solution and platelet count were calculated under the light microscope (0 h PLT counts). After that, 1 g/kg, 0.33 g/kg and 0.12 g/kg dose of IVIG (GC pharma) and 0.33 g/kg, 0.12 g/kg and 0.033 g/kg dose of rhIgG1 Fc were injected into intraperitoneal (i.p.) cavity. After 2 h of IVIG and rhIgG1 Fc injection, anti-platelet antibody MWReg30 (0.1 µg/g) was injected into i.p cavity. After 4 h of MWReg30, blood were extracted from tail vein and platelet staining was performed using 1% brilliant cresyl blue solution, and platelets counts were calculated under the light microscope (4 h PLT counts). Percentage of residual platelet counts were calculated dividing 4 h PLT counts by 0 h PLT counts.

**Flow cytometry analysis.** The spleen of each mouse was harvested at 6 h after PBS, IVIG (1 g/kg) and rhIgG1 Fc (0.33 g/kg) injection. The spleen cells were stained using a mouse Treg Detection Kit (130-120-674, Miltenyi Biotic, Bergisch Gladbach, Germany). The cells were stained with APC labeled antimouse CD25 antibody and Vio-Blue-labeled CD4 antibody for 30 min under refrigeration. The cells were then washed with PBS, permeabilized, and fixed using a fixation/permeabilization solution for 30 min in dark under refrigeration. After washing twice, the cells were treated with PE-labeled anti-mouse Foxp3 antibody for 30 min in dark under refrigeration. The stained cells were run on a flow cytometer (BD Biosciences, San Jose, CA) and data were analyzed using a FlowJo software (Tree Star, Ashland, OR).

**Analysis of gene expression by RT-qPCR.** The spleen of each mouse was harvested at 6 h after PBS, IVIG (1 g/kg) and rhIgG1 Fc (0.33 g/kg) injection. Total RNA was extracted from spleen samples using TRIzol reagent (Molecular Research Center, USA) in accordance with the manufacturer's protocol and the cDNA was synthesized using the Superscript III First-Strand Synthesis System (Invitrogen). The PCR mixture contained 2 µL of PCR buffer, 1 µL of 20× EvaGreen qPCR dye (Biotium, Hayward, CA, USA), 0.4 µL of 10 mmol/L dNTP mixture, and 10 pmol each of gene-specific forward and reverse primers (Supplementary Table 3). RT-qPCR were performed in triplicate. Relative target gene expression was quantified after normalization against mouse glyceraldehyde 3-phosphate dehydrogenase (GAPDH) expression as an endogenous control.

**In vivo serum half-life analysis.** For half-life analysis, rhIgG1 Fc purified from egg yolk and recombinant Fc derived from HEK293 cell (10702-HNAH, Sino Biological Inc) were i.p. injected into 8 weeks of female C57BL/6 mice at dose of 100 µg/mouse. After 24 h of injection, blood was collected via tail vein for 5 days and serum was obtained via centrifugation. The serum concentration of rhIgG1 Fc purified from egg yolk, and recombinant Fc derived from HEK293 cell was determined by ELISA using a human IgG ELISA kit (Abcam). Half-life was calculated by nonlinear regression analysis.

**Statistics and reproducibility.** Each experiments were performed with at least three biologically independent samples and animals for reproducibility and statistical analysis was performed using GraphPad Prism (GraphPad Software, La Jolla, CA, USA). Significant differences between more than two groups were determined by a one-way ANOVA analysis of variance with Bonferroni's multiple comparison test. Statistical analysis between two groups was analyzed by a paired *t*-test. The data were presented as mean ± standard deviation values. A value of $P < 0.05$ indicated statistical significance.

**Reporting summary**. Further information on research design is available in the Nature Portfolio Reporting Summary linked to this article.

## Data availability

The datasets generated during and/or analyzed during the current study can be found in the figures, tables and supplementary information. Source data used to generate graphs in the main text are available in Supplementary Data 1. Unprocessed gel image can be found in Supplementary Fig. 5. All other data are available from the corresponding author on reasonable request.

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

## Acknowledgements

This work was supported by the National Research Foundation of Korea (NRF) grant funded by the Korea government (MSIT) [NRF-2015R1A3A2033826].

## Author contributions

J.Y.H. participated in study design and overall coordination; J.S.P participated in study design and carried out the experiments, data interpretation and wrote the first draft of the manuscript; H.J.C. carried out the experiments, data interpretation and wrote the first draft of the manuscript. K.M.J, K.Y.L and J.H.S. carried out experiments. K.J.P carried out experimental animal management. Y.M.K. and J.Y.H. carried out data interpretation and involved in writing of final version of manuscript.

## Competing interests

The authors declare no competing interests.
