## [Peer Review File · Communications Biology]

Reviewers' comments:

Reviewer #1 (Remarks to the Author):

This manuscript describes the production of the human IgG1 Fc fragment in transgenic chickens for the purpose of manufacturing a replacement for high-dose human-derived IVIG therapy. The expressed protein was purified from chicken serum or egg yolk and characterized extensively for its glycosylation, binding to Fc receptors, and activity in an in vivo model. The conclusion is that expression in the chicken results in low-fucosylation which increases the affinity of the Fc to certain Fc receptors and therefore increases activity in ADCC. The manuscript is well-written, and the study is well-designed and clearly explained for the most part. My main question is whether expression of human Fc in chickens is really necessary to solve a problem of supply of IVIG. The claim is made that the transgenic chicken approach would be more cost-effective and provide a more reliable supply, but data are lacking on that idea. If a recombinant approach is desired, how does the chicken system compare to expression in fucosyltransferase-deficient CHO cells? Could we have comparison of the costs of these approaches relative to that of IVIG sourced from humans?

Questions on some of the details:

Data on germline transmission would be appreciated. How many chimeras were made, how many offspring were screened for transmission, etc.

Are homozygous animals healthy? Do they produce normal levels of albumin? Do the hens lay eggs? Only serum up to 8 weeks was analyzed in homozygotes, not egg yolk, implying homozygous hens don't lay eggs.

How pure are the preparations of rh Fc? Is there any chicken IgY or albumin-Fc? The Coomassie gel on Fig 3a looks like the purity is high but the mass spec data in S2 and S3 are confusing. Tables in S2 and S3 should be described, there is no legend.

What is the half-life of the recombinant hFc in mice?

Could you comment on why such a vast excess (1000-fold) of IVIG or Fc is needed to block ADCC activity or anti-platelet activity in the in vivo model?

Figure 5d seems to be missing some labeling. The scatter plots should be labeled with what sample is analyzed on each plot.

Figure S2. There needs to be a positive control for DC-SIGN binding, otherwise the two negative results from IVIG and Fc are not meaningful. Especially because this was an unexpected result.

Reviewer #3 (Remarks to the Author):

Summary

Park et al. address the point of IVIG being expensive and of limited supply, by suggesting an alternative IgG1 Fc alternative. This was generated using genome-edited chickens, with CRISPR/Cas9-NHEJ, that produce recombinant human IgG-Fc (rhIgG1) in serum and egg yolk. In addition, they show, with LC/MC/MC, that these chickens produce rhIgG1 with high sialylation and low fucosylation levels, features that they show to exert anti-inflammatory effects in an in vitro ADCC assay and a passive ITP mouse model.

The manuscript is written in a concise manner and the data is presented in a logical order. However,

the message of the manuscript is oversold and more evidence is needed to support certain statements. The following points of concern can be raised regarding the scientific content of the manuscript:

Major points

- NHEJ donor plasmid: For the CRISPR/Cas9-NHEJ design, a T2A tag was used in the donor plasmid in order to separate the ALB protein from the rhIgG1 Fc, but the function, rationale and potential limitations of this tag is not explained. E.g. why is it not fully cleaved?
- It is stated several times that that production of rhIgG1 Fc in chickens can be an alternative source of IVIG that reduces the costs. This seems to be an unlikely approach towards FDA approved product, and that is not taking into account that the concentration in blood and yolk is rather low, making this non-feasible.
- The anti-inflammatory effects of sialylated IgG-Fc and also DC-SIGN as receptor for IgG are highly debatable and been a topic of a body of literature suggesting previous papers claiming DC-SIGN to be a receptor for human IgG to be faulty. Results you present and discuss in line 225 are in accordance with that recently published by Temming et al Scientific reports 9.1 (2019): 1-10) suggesting DC-SIGN does not bind human IgG. This needs to be mentioned.
- L268. The in vivo mouse model is not introduced in the result section. Explain the model and cite an original reference for this (e.g. Blood. 2001;98(4):1095-1099).
- L156, L159 and Fig2b. In the text, a band of 50kDa is mentioned and this is also observed in the figure. However, the text next to the band in the figure states 70kDa. Moreover, the reduced band is 35, and the authors explain that this is the size of glycosylated CH2+CH3. But is the non-reduced 50kDa band then deglycosylated?
- The authors performed an in vitro ADCC experiments with FcγRIIIa-expressing Jurkat cells. Jurkat cells are T cell-derived that do not express FcγR (<https://www.ncbi.nlm.nih.gov/pmc/articles/PMC3994145/>). Where did the FcγRIII originate, were these cells generated or were these WT Jurkat?
- The effector cells used for the ADCP assay are not described nor the principles of the assay. It is also very surprising that IVIG nor the Fc fragment has an effect. This has been described by numerous other papers.
- The main text is often written in large paragraphs. Please try to be a little more concise and split up those large paragraphs in smaller logical units to improve readability.

Minor points

- Explain a little better in the introduction that afucosylation of antigen-specific IgG is proinflammatory, but bulk antigen-specific IgG can be anti-inflammatory due to blocking of FcγRIIIa. Two papers actually have shown that FcγRIIIa are preferentially occupied by aspecific afucosylated IgG1 in humans in vivo. (<https://pubmed.ncbi.nlm.nih.gov/31748349/> and DOI:10.1074/mcp.RA119.001607).
- ?
- L30. Spell out ITP at first appearance as immune thrombocytopenia
- T94. 'do not produce non-human glycans'. Difficult sentence, could be changed into 'only produce human glycans' or something similar.
- L97. Difficult sentence, write down differently.
- L113. Write rhIgG1 directly after 'The recombinant human IgG1 Fc.....'
- L115. Change 'chicken' to 'chickens'. This also applies for L121.
- L138. Fig. 1e is written there but should be removed.
- L145. Explain why G1, G2 and G3 progeny are needed. As heterozygous G2 is mated resulting in homozygous G3, the question is whether G3 is also fertile? Is there also reduced albumin expression in G2 and G3 and does this have consequences? Moreover, in the next experiments, are the G2 or G3 used? This is unclear from the text.
- L153. 'expressed' needs to be 'transcribed' as we are talking about RNA.
- L154. ALB::hIgG1 Fc is ONLY transcribed in the liver. This is not the organ expressing the highest amount, as the other organs don't show the presence of ALB::hIgG1 Fc RNA.

- L168. An error bar of +/- 71,65 is mentioned in the text, but the error bar in the corresponding figure (Fig 2c) is very small.
- L175. 'generation' is confusing, rather use progenies or something similar.
- L194. A comparison is made between ALB::hIgG1 Fc glycosylation and human glycosylation patterns, but this is not clearly explained. In addition, use proper reference and mention the IgG glycosylation profile observed in human serum.
- L205. Add the word 'the' in 'is largely THE same as'. This also applies for L207 'to identify THE major linkage'.
- L233. As this result section is based on SPR data, the conclusion that rhIgG1 Fc has higher affinity for FcγRIIIa can be made, but the conclusion that it also has higher blocking activity does not fit here.
- L277. Anti-inflammatory activity is mentioned here, but this is rather blocking activity. It would be more clear if these two definitions are more strictly separated in the last two result sections.
- L307. The word efficient is mentioned two times.
- L310 and L313. Try to find a different word for 'also' in one of the sentences. This also applies for L355 and L357 where the word suggested is used twice.
- L357. Make this message more clear. What is meant by human blood products? IgG or also other proteins?
- Fig1b. Explain DW
- L596. Add the word 'for' after 'using primers specific FOR...'
- Fig3e. It would be nice to have a positive control for the MAL II blot.
- Fig5d. Mention PBS, IVIG and rhIgG1 conditions in the FACS dotplots as they are lacking.
- Supplementary figure 2: the scale and overall layout for the Fc and binding to FcγRII seems off- all the points seem on the Y axes at 0 M (Is the unit of the X axes correct?) with a random line protruding from the X at ca 3.3 M.
- Eggs, normally containing rather large amounts of IgY antibodies, are consumed. It would be interesting to see in future projects how well the Fc fragments survive the gastroenteric tract and if they are taken up by FcRn in the gut, starting with mouse models.

Response to Reviewer #1

This manuscript describes the production of the human IgG1 Fc fragment in transgenic chickens for the purpose of manufacturing a replacement for high-dose human-derived IVIG therapy. The expressed protein was purified from chicken serum or egg yolk and characterized extensively for its glycosylation, binding to Fc receptors, and activity in an in vivo model. The conclusion is that expression in the chicken results in low-fucosylation which increases the affinity of the Fc to certain Fc receptors and therefore increases activity in ADCC. The manuscript is well-written, and the study is well-designed and clearly explained for the most part.

My main question is whether expression of human Fc in chickens is really necessary to solve a problem of supply of IVIG. The claim is made that the transgenic chicken approach would be more cost-effective and provide a more reliable supply, but data are lacking on that idea. If a recombinant approach is desired, how does the chicken system compare to expression in fucosyltransferase-deficient CHO cells? Could we have comparison of the costs of these approaches relative to that of IVIG sourced from humans?

Author's response : Thank you for your kind evaluation on our manuscript. The main purpose of present manuscript is to establish bioreactor system that express target protein in liver specific manner and eventually accumulate target protein at egg yolk, with abundant α -2,6 sialylated and afucosylated glycosylation pattern. In present study, IgG1 Fc can be accumulated as 3-4 mg/egg and because chicken eggs can be produced with cost-effectiveness (around 10 cents per egg), we suggested here that chicken bioreactor can be cost-effective system for producing human IgG1 Fc. Although we do not directly compare with fucosyltransferase-deficient CHO cells here, chicken bioreactors are generally recognized as cost-effective methods than mammalian cell culture system (*Zhu et al., Nat Biotechnol, 2005 ; Lillico et al., Drug Discov Today, 2005*). Meanwhile, we agreed to your points that more improved efficiency of human Fc production in eggs should be required to solve the problem of supply of IVIG and we will continuously research to improve efficiency of our system. Based on your review point, we have revised manuscript in general to reduce our description on the cost-effectiveness of our system (**Line 30-31, Line 114, Line 308-309 in the revised manuscript**) and we changed title of manuscript “*Human IVIG alternative with beneficial N-glycosylation pattern for anti-inflammatory activity derived from genome edited chickens*” to “*Production of recombinant human IgG1 Fc with beneficial N-glycosylation pattern for anti-inflammatory activity using genome edited chickens*”. We hope our revisions could satisfy you.

Reviewer's comments : Data on germline transmission would be appreciated. How many chimeras were made, how many offspring were screened for transmission, etc.

Author's response : We add table on the information of germline transmission in

Supplementary Table S1 and add description **L137-138** in the revised manuscript.

Reviewer's comments : Are homozygous animals healthy? Do they produce normal levels of albumin? Do the hens lay eggs? Only serum up to 8 weeks was analyzed in homozygotes, not egg yolk, implying homozygous hens don't lay eggs.

Author's response : When we prepared the first draft of manuscript, we have homozygous animals only up to 8 weeks, so we cannot add data on homozygote egg yolk. The homozygous animals are healthy and able to lay eggs after sexual maturation. We add data on accumulation of rhIgG1 Fc in homozygous serum and egg yolk in **Supplementary Figure S2**. Also, for identifying that whether homozygous animals produce albumin normally, we performed SDS-PAGE and Coomassie blue staining of serum of three homozygous and wild type hens and observed that albumin was also secreted into blood in homozygous animals (**Supplementary Figure S2, Line 146-149** in the revised manuscript).

Reviewer's comments : How pure are the preparations of rhFc? Is there any chicken IgY or albumin-Fc? The Coomassie gel on Fig 3a looks like the purity is high but the mass spec data in S2 and S3 are confusing. Tables in S2 and S3 should be described, there is no legend.

Author's response : We have purified hIgG1 Fc using protein A column and size exclusion chromatography and confirm purification of hIgG1 Fc in SDS-PAGE. Although the exact purity of sample can be determined by HPLC, we didn't further analysis because it was identified in SDS-PAGE that purity is high enough for conducting further experiments. Additionally, because protein A does not bind to chicken IgY (*Ansari & Chang., Am J Vet Res, 1983; W.W. Zhang., Drug Discov Today, 2003*), we carefully expect that IgY may not be included in our purified hIgG1 Fc samples. In LC/MS/MS, we crop the band of rhIgG1 Fc in SDS-PAGE gel and extract protein, degrade into several peptides, and conduct mass spectrometry. Because degraded peptide fragments are analyzed, not a full protein sequence, there is possibility to annotate unrelated proteins. Although we identified that the annotated proteins with high coverage have amino acid sequences of hIgG1 Fc (NCBI Accession number Q6N096, Q86TT2, A0A286YFY4, Q6MZX7), the name of proteins associated with accession number are differed from hIgG1 Fc, which may bring confusion to readers. Therefore, for reducing confusions, we suggest that it is better to remove supplementary tables on the LC/MS/MS and we have revised our manuscript. We hope our revision could be acceptable for you.

Reviewer's comments : What is the half-life of the recombinant hFc in mice?

Author's response : According to your comment, we have injected rhIgG1 Fc derived from yolks and recombinant Fc produced from HEK293 cells (Cat No. 10702-HNAH, SinoBiological) into C57BL/6 female mouse and measure serum concentrations of rhIgG1 Fc

during several days after injection for measuring half-life. The half-life of rhIgG1 Fc derived from yolks and recombinant Fc produced from HEK293 cells was measured as 39.14 and 36.37 hours, respectively. Please see **Supplementary Fig S3** and **L220-223 in the revised manuscript**.

Reviewer's comments : Could you comment on why such a vast excess (1000-fold) of IVIG or Fc is needed to block ADCC activity or anti-platelet activity in the in vivo model?

Author's response : IVIG is polyclonal IgG antibodies purified from pooled human plasma of thousands of people's blood. Initially, IVIG was used to confer passive immunity to patients with compromised immunity. Meanwhile, it was discovered that when IVIG was administered at high dose (1-2g/kg), it induced anti-inflammatory response and restore platelet counts in ITP patients (*Imbach et al., Lancet, 1981*). Thereafter, high dose IVIG treatments have been widely used in treatment of various inflammatory and autoimmune diseases. Although the exact mechanisms of anti-inflammatory activity of high-dose IVIG have not been clearly elucidated yet, there are several suggested mechanisms such as sialylated Fc mediated anti-inflammatory activities and inhibition of autoantibody-antigen immune complexes binding to Fc receptors by competitive blockade of Fc receptors. Because only 10% of IgG have sialylated Fc in IVIG, it was suggested that vast excess of IVIG should be used to induce anti-inflammatory activities and several studies have shown that enrichment of sialylated IgG can induce anti-inflammatory activities at lower doses (*Kaneko et al., Science, 2006*). In the aspect of Fc receptor blockade, monovalent binding affinity of IgG to FcγRII and FcγRIII is very low. Therefore, vast excess of IVIG or Fc is required to competitively block binding of auto-immune complexes to these Fc receptors and induce anti-inflammatory activities (*Nagelkerke & Kuijpers., Front Immunol, 2015*). We hope our explanation could satisfy you.

Reviewer's comments : Figure 5d seems to be missing some labeling. The scatter plots should be labeled with what sample is analyzed on each plot.

Author's response : Thank you for your indication of our mistake. We add missing label on scatter plot of **Figure 5d**.

Reviewer's comments : Figure S2. There needs to be a positive control for DC-SIGN binding, otherwise the two negative results from IVIG and Fc are not meaningful. Especially because this was an unexpected result.

Author's response : Thank you for your comment. Although it has been suggested that DC-SIGN can bind to sialylated Fc region of IgG, demonstrated by cell-based ELISA method (*Anthony et al., PNAS, 2008. doi: 10.1073/pnas.0810163105, Sondermann et al., PNAS, 2013. doi: 10.1073/pnas.1307864110*), several other reports using FACS or SPR method showed that DC-SIGN did not bind to Fc region of IgG (*Temming et al., Sci Rep, 2019. doi:*

10.1038/s41598-019-46484-2, Zhang et al., JCI Insight, 2019. doi: 10.1172/jci.insight.121905).
Although we don't have positive control for DC-SIGN binding as you pointed, we used SPR method to measure affinity of Fc to several Fc receptors including DC-SIGN and we expect that our data could be interpreted as in accordance with above two reports. We hope our explanation could satisfy you.

Response to Reviewer #2

Reviewer's comments : Park et al. address the point of IVIG being expensive and of limited supply, by suggesting an alternative IgG1 Fc alternative. This was generated using genome-edited chickens, with CRISPR/Cas9-NHEJ, that produce recombinant human IgG-Fc (rhIgG1) in serum and egg yolk. In addition, they show, with LC/MC/MC, that these chickens produce rhIgG1 with high sialylation and low fucosylation levels, features that they show to exert anti-inflammatory effects in an in vitro ADCC assay and a passive ITP mouse model.

The manuscript is written in a concise manner and the data is presented in a logical order. However, the message of the manuscript is oversold and more evidence is needed to support certain statements. The following points of concern can be raised regarding the scientific content of the manuscript

Author's response : The authors are thankful to this reviewer for provided a positive evaluation on our manuscript, and several comments that greatly improved the manuscript. The manuscript was revised extensively and also prepared a point-by-point response to your comments. Therefore, we believe that the revised manuscript could satisfy your point of view on our manuscript. Please see below for our responses on your specific comments, and the manuscript for the corresponding revision.

1. Major Points

Reviewer's comments : NHEJ donor plasmid: For the CRISPR/Cas9-NHEJ design, a T2A tag was used in the donor plasmid in order to separate the ALB protein from the rhIgG1 Fc, but the function, rational and potential limitations of this tag is not explained. E.g. why is it not fully cleaved?

Author's response : To prevent side effects that can be caused by albumin deficient, we linked rhIgG1 Fc to albumin using T2A self-cleavage peptide and we intended to separate rhIgG1 Fc from albumin, although cleavage efficiency of 2A peptides cannot be reached to 100%. We describe the rational of using T2A tag in **Line 125-130 in the revised manuscript** according to your comments.

Reviewer's comments : It is stated several times that that production of rhIgG1 Fc in chickens can be an alternative source of IVIG that reduces the costs. This seems to be unlikely approach towards FDA approved product, and that is not taking into account that the concentration in blood and yolk is rather low, making this non-feasible.

Author's response : Thank you for your comment to improve quality of our manuscript. During preparation of manuscript, we intended to suggest that chicken liver specific expression system can produce recombinant proteins with higher sialylation and lower fucosylation ratio, and this glycosylation patterns could be beneficial to anti-inflammatory activity of IgG1 Fc. Based on this idea, we think chicken can be one of potential alternative source of human IVIG as anti-inflammatory agents. In present study, IgG1 Fc can be accumulated as 3-4 mg/egg and because chicken eggs can be produced with cost-effectiveness (around 10 cents per egg), we suggested here that chicken bioreactor can be cost-effective system for producing human IgG1 Fc. However, we agreed to your points that more improved efficiency of human Fc production in eggs should be required to solve the problem of supply of IVIG and we will continuously research to improve efficiency of our system. According to your comment, we have revised manuscript in general to reduce our description on the cost-effectiveness of our system (**Line 30-31, Line 114, Line 308-309 in the revised manuscript**) and we changed title of manuscript "*Human IVIG alternative with beneficial N-glycosylation pattern for anti-inflammatory activity derived from genome edited chickens*" to "*Production of recombinant human IgG1 Fc with beneficial N-glycosylation pattern for anti-inflammatory activity using genome edited chickens*".

Reviewer's comments : The anti-inflammatory effects of sialylated IgG-Fc and also DC-SIGN as receptor for IgG are highly debatable and been a topic of a body of literature suggesting previous papers claiming DC-SIGN to be a receptor for human IgG to be faulty. Results you present and discuss in in line 225 are in accordance with that recently published by Temming et al Scientific reports 9.1 (2019): 1-10) suggesting DC-SIGN does not bind human IgG. This needs to be mentioned.

Author's response : Thank you for your comments. We have mentioned above reference about the DC-SIGN was not bona fide receptor for human IgG1 Fc, as you recommended in discussion section **Line 350-356 in the revised manuscript**.

Reviewer's comments : L268. The in vivo mouse model is not introduced in the result section. Explain the model and cite an original reference for this (e.g. Blood. 2001;98(4):1095-1099).

Author's response : According to your comment, we introduce in vivo mouse model and cite original reference for this model as you recommended (**Line 279-281 in the revised manuscript**).

Reviewer's comments : L156 and L159 and Fig2b. In the text, a band of 50kDa is mentioned and this is also observed in the figure. However, the text next to the band in the figure states 70kDa. Moreover, de reducing band is 35, and the authors explain that this is the size of glycosylated CH2+CH3. But is the non-reduced 50kDa band then deglycosylated?

Author's response : As you pointed out, we found that our description is rather confusing to readers. We have not performed any deglycosylation in our western blot experiment in Figure 2b. In reducing conditions, disulfide bonds are cleaved and the proteins are linearized and localized to its exact molecular weight band size when SDS-PAGE gel running. However, in non-reducing conditions, the proteins still folded into 3D structure and because of this nature, the band size of target proteins tends to be down-shifted than expected molecular weight. To prevent confusion, we deleted band size description in non-reducing condition in manuscript (**Line 160-161 in the revised manuscript**) and in **Figure 2B**.

Reviewer's comments : The authors performed an in vitro ADCC experiments with FcγRIIIa-expressing Jurkat cells. Jurkat cells are T cell-derived that do not express FcγR (<https://www.ncbi.nlm.nih.gov/pmc/articles/PMC3994145/>). Where did the FcγRIII originate, were these cells generate bought or were these WT Jurkat?

Author's response : In ADCC experiments, we used ADCC Reporter Bioassay, V variant complete kit (WIL2-S) manufactured by Promega (Cat No. G7014). The transgenic Jurkat cells that express human FcγRIIIA (V158) was provided in this kit and used as effector cells. These transgenic Jurkat cells also have been engineered to induce luciferase activity when NFAT pathway is activated by FcγRIIIA crosslinking. We have incubated these effector cells with serially diluted anti-CD20 antibody, WIL2-S target cells that express CD20, and IVIG or rhIgG1 Fc and examine the level of ADCC induction (**Line 250-256 and Methods section (Line 532-541) in the revised manuscript**).

Reviewer's comments : The effector cells used for the ADCP assay are not described nor the principles of the assay. Its also very surprising that IVIg nor the Fc fragment has an effect. This has been described by numerous other papers.

Author's response : In ADCP assay, we used FcγRIIA-H ADCP Bioassay kit manufactured by Promega (Cat No. G9901) and assay principles are same with ADCC assay. We have described ADCP measuring principles more detail in **Line 268-276**. Although ADCC was effectively inhibited in 1.8 mg/ml of IVIG and 0.6 mg/ml of rhIgG1 Fc, ADCP was not effectively inhibited in same concentration of IVIG and rhIgG1 Fc. This may be resulted from relatively lower affinity of IVIG and rhIgG1 Fc to FcγRIIA than FcγRIIIA and more higher concentration of IVIG and rhIgG1 Fc will be required to have blocking ability to FcγRIIA in our experimental setting. We have changed our description as FcγRIIA blocking activity cannot be observed at concentration of 1.8 mg/ml of IVIG and 0.6 mg/ml of Fc (**Line 268-276 in the revised manuscript**). Also, we describe assay method more specifically in **Materials and**

methods section. We hope our explanation could be acceptable for your standard.

Reviewer's comments : The main text is often written in large paragraphs. Please try to be a little more concise and split up those large paragraphs in smaller logical units to improve readability.

Author's response : We appreciate for your comment to improve quality of our manuscript. We revised large paragraphs into more smaller units (**Line 61-62, 76-77, 130-131, 142-143, 157-158, 169-170 in the revised manuscript**) according to your recommendation.

2. Minor Points

Reviewer's comments: Explain a little better in the introduction that afucosylation of antigen-specific IgG is proinflammatory, but bulk antigen-aspecific IgG can be anti-inflammatory due to blocking of FcγRIIIa. Two papers actually have shown that FcγRIIIa are preferentially occupied by aspecific afucosylated IgG1 in humans in vivo. (<https://pubmed.ncbi.nlm.nih.gov/31748349/> and DOI:10.1074/mcp.RA119.001607).

Author's response: According to your comments, we further explain afucosylated antigen-aspecific antibody can be anti-inflammatory and cited references of your recommendation (**Line 72-76 in the revised manuscript**).

Reviewer's comments: L30. Spell out ITP at first appearance as immune thrombocytopenia

Author's response: We spell out ITP as immune thrombocytopenia according to your recommendation.

Reviewer's comments: T94. 'do not produce non-human glycans'. Difficult sentence, could be changed into 'only produce human glycans' or something similar.

Author's response: We have changed our description according to your recommendation (**Line 90 in the revised manuscript**).

Reviewer's comments: L97. Difficult sentence, write down differently.

Author's response: We have changed our description for improving readability according to your recommendation (**Line 92-94 in the revised manuscript**)

Reviewer's comments: L113. Write rhIgG1 directly after 'The recombinant human IgG1 Fc.....'

Author's response: We corrected main text according to your recommendation (**Line 110**).

Reviewer's comments: L115. Change 'chicken' to 'chickens'. This also applies for L121.

Author's response: We corrected words according to your recommendation (**Line 113 and Line 118**).

Reviewer's comments: L138. Fig. 1e is written there but should be removed.

Author's response: We corrected text according to your recommendation (**Line 140**).

Reviewer's comments: L145. Explain why G1, G2 and G3 progeny are needed. As heterozygous G2 is mated resulting in homozygous G3, the question is whether G3 is also fertile? Is there also reduced albumin expression in G2 and G3 and does this have consequences? Moreover, in the next experiments, are the G2 or G3 used? This is unclear from the text.

Author's response: In G1, we obtained only one *ALB::hIgG1 Fc* rooster that finally reached sexual maturation. Therefore, we perform mating between G1 *ALB::hIgG1 Fc* rooster and wild type hen, producing G2 heterozygous *ALB::hIgG1 Fc* progenies. By mating between G2 progenies, we produced G3 progenies with homozygous for *ALB::hIgG1*. The homozygous *ALB::hIgG1* chickens can reach sexual maturation and lay eggs, which means homozygous *ALB::hIgG1* chickens are also fertile. rhIgG1 Fc also accumulated in egg yolk of eggs laid by homozygous *ALB::hIgG1* chickens. We add data on the analysis of serum and egg yolk of homozygous *ALB::hIgG1* chickens in **Supplementary Figure S2**. And also please see **L146-150**. We hope our revision could satisfy you.

Reviewer's comments: L153. 'expressed' needs to be 'transcribed' as we are talking about RNA.

Author's response: We corrected 'expressed' into 'transcribed' as your recommendation. Please see **L157**.

Reviewer's comments: L154. ALB::hIgG1 Fc is ONLY transcribed in the liver. This is not the organ expressing the highest amount, as the other organs don't show the presence of ALB::hIgG1 Fc RNA.

Author's response: We changed our description as "rhIgG1 Fc was transcribed successfully in the liver specific manner" (**Line 156-157**)

Reviewer's comments: L168. An error bar of +/- 71,65 is mentioned in the text, but the error bar in the corresponding figure (Fig 2c) is very small.

Author's response: We corrected our errors. Please see **Fig 2c**.

Reviewer's comments: L175. 'generation' is confusing, rather use progenies or something similar.

Author's response: We corrected main text as your recommendation and change "generation" to "progenies". Please see **L177**.

Reviewer's comments: L194. A comparison is made between ALB::hIgG1 Fc glycosylation and human glycosylation patterns, but this is not clearly explained. In addition, use proper reference and mention the IgG glycosylation profile observed in human serum.

Author's response: We revised our manuscript according to your comments with reference describing N-glycosylation pattern of human serum proteins. Please see **L372-377 of revised manuscript**.

Reviewer's comments: L205. Add the word 'the' in 'is largely THE same as'. This also applies for L207 'to identify THE major linkage'.

Author's response: We added 'the' according to your recommendation. Please see **L209 and L211**.

Reviewer's comments: L233. As this result section is based on SPR data, the conclusion that rhIgG1 Fc has higher affinity for FcγRIIIa can be made, but the conclusion that it also has higher blocking activity does not fit here.

Author's response: We revised manuscript according to your comment. Please see **L240-241**.

Reviewer's comments: L277. Anti-inflammatory activity is mentioned here, but this is rather blocking activity. It would be more clear of these two definitions are more strictly separated in

the last two result sections.

Author's response: We changed our description according to your recommendation. Please see **L279-291**

Reviewer's comments: L307. The word efficient is mentioned two times.

Author's response: We revised 'efficient sialylation efficiency' to 'efficient sialylation ratio' Please see **L317**.

Reviewer's comments: L310 and L313. Try to find a different word for 'also' in one of the sentences. This also applies for L355 and L357 where the word suggested is used twice.

Author's response: We changed 'Also' into 'Additionally' in **L323** and changed 'suggested' into 'proposed' in **L372**.

Reviewer's comments: L357. Make this message more clear. What is meant by human blood products? IgG or also other proteins?

Author's response: In this statement, we intended to suggest that although it is not fully demonstrated yet, chicken liver bioreactor can be one of optimal production platform for human blood products that are synthesized from human liver such as blood clotting factors and alpha-1 antitrypsin because glycosylation pattern of liver derived proteins from these two species is similar. We tried to clarify the message by revising our description at **L372-377** according to your recommendation.

Reviewer's comments: Fig1b. Explain DW

Author's response: We changed term DW (distilled water) to ddH₂O. Please see Fig 1b.

Reviewer's comments: L596. Add the word 'for' after 'using primers specific FOR...'

Author's response: We corrected legend of Figure 1 according to your comments.

Reviewer's comments: Fig3e. It would be nice to have a positive control for the MAL II blot.

Author's response: According to your recommendation, we used recombinant EPO derived from CHO cells (Cat No. 100-64, Peprotech) as positive control for the MAL II blot and revised figure. Please See Fig3e of the revised manuscript.

Reviewer's comments: Fig5d. Mention PBS, IVIG and rhIgG1 conditions in the FACS

dotplots as they are lacking.

Author's response: Thank you for your indication of our mistake. We add missing label on scatter plot of Figure 5d.

Reviewer's comments: Supplementary figure 2: the scale and overall layout for the Fc and binding do Fc γ RII seems off– all the points seem on the Y axes at 0 M (Is the unit of the X axes correct?) with a random line protruding from the X at ca 3.3 M.

Author's response: As you mentioned, in our SPR experiment, we cannot detect any binding between rhIgG1 Fc and Fc γ RII in all range of concentration. The values plotted on the sensorgram and units for both X and Y axis are automatically analyzed by BiaEvaluation 3.01 software and we used sensorgram without any modifications.

Reviewer's comments: Eggs, normally containing rather large amounts of IgY antibodies, are consumed. It would be interesting to see in future projects how well the Fc fragments survive the gastro enteric tract and if they are taken up by FcRn in the gut, starting with mouse models.

Author's response: Thank you for your suggestion on the future research project and give us opportunity for studying FcRn. Based on your suggestion, we will continue to research on the delivery of yolk Fc into gastro enteric tract or nasal cavity to apply development of edible vaccines or bio-drugs formulated by egg yolk.

Reviewers' comments:

Reviewer #1 (Remarks to the Author):

Thank you for addressing my comments and making improvements to the manuscript. I still have a few lingering concerns.

Typo in the new title (in the PDF version): ..."using from genome edited chickens" should be changed to either "using" or "from"

Lines 139, 141, 177, 435: "progeny" is both singular and plural (i.e. don't use "progenies")

Line 177, not sure what is meant here by progeny. Generation? Zygosity? Genetic lineage from original PGCs?

Comments on breeding:

Mendelian inheritance should be confirmed in the breeding to homozygosity (i.e. 1:2:1 ratio of genotypes in progeny of heterozygous matings), in S1.

The chimera test mating needs to be better explained. According to the legend, the transplanted donor PGCs are I/I, the KO recipients of the PGCs are I/i and the wild type hens used for mating to the chimeras are I/I. Thus both donor-derived and recipient-derived progeny could be I/I, and half of the recipient-derived progeny would be I/i. Thus to calculate frequency of germline transmission you are looking for loss of I/i, not gain of a specific genotype, which makes it much less useful than if a specific genotype is associated with germline transmission.

Figure S2: calculation of ALB concentration in serum and/or eggs from WT, het and hom birds would be much more informative than the Coomassie gel only showing WT and homozygous

Line 147: sentence is not right: "We observed that homozygous chickens also secrete ALB into blood and healthy to have sexual maturation, and lay eggs" should be changed to something like "We observed that homozygous chickens also secrete ALB into blood, are healthy, reach sexual maturity, and lay eggs"

Line 151: "maintained as a homozygous breed" has not been shown; so far, you have shown that eggs are laid by homozygous females but not that they would produce viable offspring. You either need to show hatching and rearing of chicks from homozygous parents, or remove the statement.

Reviewer #3 (Remarks to the Author):

The authors answered all reviewers comments and the concerns raised were resolved satisfactorily. This reviewer considers the manuscript in an acceptable form for publication.

Response to Reviewer #1

We appreciate to Reviewer #1 for providing detailed review and valuable comments that help us to improve quality of our manuscript. Based on your comments, we revised our manuscript and we hope that our revisions could be acceptable for you.

Reviewer's comment : Typo in the new title (in the PDF version): ...“using from genome edited chickens” should be changed to either “using” or “from”.

Author's response : We revised the title as ...“using genome edited chickens”. Please see Line 2.

Reviewer's comment : Lines 139, 141, 177, 435: "progeny" is both singular and plural (i.e. don't use “progenies”)

Author's response : We corrected words. Please see Lines 139, 141, 178, 435.

Reviewer's comment : Line 177, not sure what is meant here by progeny. Generation? Zygosity? Genetic lineage from original PGCs?

Author's response : In here, we intended to describe that Fc could be continuously secreted into bloodstream during several generations without transgene silencing. We revised description for more clear delivery as “regardless of progeny in several generations” Please see Line 178.

Reviewer's comment : Mendelian inheritance should be confirmed in the breeding to homozygosity (i.e. 1:2:1 ratio of genotypes in progeny of heterozygous matings), in S1.

Author's response : According to your comments, we have analyzed genotype of chicks hatched from heterozygous mating. From analyzed 18 chicks, we obtained four wild type, nine heterozygous and five homozygous chicks, which approximately follow 1:2:1 ratio of Mendelian inheritance. Please see Figure S1. and Line 146-148.

Reviewer's comment : The chimera test mating needs to be better explained. According to the legend, the transplanted donor PGCs are I/I , the KO recipients of the PGCs are I/i and the wild type hens used for mating to the chimeras are I/I . Thus both donor-derived and recipient-derived progeny could be I/I , and half of the recipient-derived progeny would be I/i . Thus to calculate frequency of germline transmission you are looking for loss of I/i , not gain of a specific genotype, which makes it much less useful than if a specific genotype is associated with germline transmission.

Author's response : We found error in Table S1 legend. The genotype of KO recipient is i/i , not I/i . We used PGCs derived from WL (I/I) and transplanted these PGCs into KO (i/i)

recipients. Therefore, germline chimeric KO could produce sperm of *I* (derived from WL donor PGCs) and *i* (derived from KO endogenous germ cell). After mating between wild type WL (*I/I*) hen and germline chimeric KO, the donor PGC derived progeny will be *I/I* (WL) and endogenous KO germ cell derived progeny will be *I/i* (hybrid). In this regards, for calculating germline transmission efficiency, we calculated the ratio of donor PGC derived progeny (*I/I*) from total hatched chicks. We have revised legend of Table S1.

Reviewer's comment : Figure S2: calculation of ALB concentration in serum and/or eggs from WT, het and hom birds would be much more informative than the Coomassie gel only showing WT and homozygous

Author's response : According to your comments, we have calculated ALB concentration of WT, heterozygous and homozygous hens. The ALB protein secreted into bloodstream regardless of genotype although its concentration have a tendency to decrease in heterozygous and homozygous birds compared to wild type birds. Please See Figure S2 and Line 148-150.

Reviewer's comment : Line 147: sentence is not right: “We observed that homozygous chickens also secrete ALB into blood and healthy to have sexual maturation, and lay eggs” should be changed to something like “We observed that homozygous chickens also secrete ALB into blood, are healthy, reach sexual maturity, and lay eggs”

Author's response : We revised sentence according to your comments and please see Line 148-150.

Reviewer's comment : Line 151: “maintained as a homozygous breed” has not been shown; so far, you have shown that eggs are laid by homozygous females but not that they would produce viable offspring. You either need to show hatching and rearing of chicks from homozygous parents, or remove the statement.

Author's response : We agreed to your comment and removed related statement. Please see Line 151-152.

Revision #1

Response to Reviewer #1

This manuscript describes the production of the human IgG1 Fc fragment in transgenic chickens for the purpose of manufacturing a replacement for high-dose human-derived IVIG therapy. The expressed protein was purified from chicken serum or egg yolk and characterized extensively for its glycosylation, binding to Fc receptors, and activity in an in vivo model. The conclusion is that expression in the chicken results in low-fucosylation which increases the affinity of the Fc to certain Fc receptors and therefore increases activity in ADCC. The manuscript is well-written, and the study is well-designed and clearly explained for the most part.

My main question is whether expression of human Fc in chickens is really necessary to solve a problem of supply of IVIG. The claim is made that the transgenic chicken approach would be more cost-effective and provide a more reliable supply, but data are lacking on that idea. If a recombinant approach is desired, how does the chicken system compare to expression in fucosyltransferase-deficient CHO cells? Could we have comparison of the costs of these approaches relative to that of IVIG sourced from humans?

Author's response : Thank you for your kind evaluation on our manuscript. The main purpose of present manuscript is to establish bioreactor system that express target protein in liver specific manner and eventually accumulate target protein at egg yolk, with abundant α -2,6 sialylated and afucosylated glycosylation pattern. In present study, IgG1 Fc can be accumulated as 3-4 mg/egg and because chicken eggs can be produced with cost-effectiveness (around 10 cents per egg), we suggested here that chicken bioreactor can be cost-effective system for producing human IgG1 Fc. Although we do not directly compare with fucosyltransferase-deficient CHO cells here, chicken bioreactors are generally recognized as cost-effective methods than mammalian cell culture system (*Zhu et al., Nat Biotechnol, 2005 ; Lillico et al., Drug Discov Today, 2005*). Meanwhile, we agreed to your points that more improved efficiency of human Fc production in eggs should be required to solve the problem of supply of IVIG and we will continuously research to improve efficiency of our system. Based on your review point, we have revised manuscript in general to reduce our description on the cost-effectiveness of our system (**Line 30-31, Line 114, Line 308-309 in the revised manuscript**) and we changed title of manuscript “*Human IVIG alternative with beneficial N-glycosylation pattern for anti-inflammatory activity derived from genome edited chickens*” to “*Production of recombinant human IgG1 Fc with beneficial N-glycosylation pattern for anti-inflammatory activity using genome edited chickens*”. We hope our revisions could satisfy you.

Reviewer's comments : Data on germline transmission would be appreciated. How many chimeras were made, how many offspring were screened for transmission, etc.

Author's response : We add table on the information of germline transmission in **Supplementary Table S1** and add description **L137-138 in the revised manuscript**.

Reviewer's comments : Are homozygous animals healthy? Do they produce normal levels of albumin? Do the hens lay eggs? Only serum up to 8 weeks was analyzed in homozygotes, not egg yolk, implying homozygous hens don't lay eggs.

Author's response : When we prepared the first draft of manuscript, we have homozygous animals only up to 8 weeks, so we cannot add data on homozygote egg yolk. The homozygous animals are healthy and able to lay eggs after sexual maturation. We add data on accumulation of rhIgG1 Fc in homozygous serum and egg yolk in **Supplementary Figure S2**. Also, for identifying that whether homozygous animals produce albumin normally, we performed SDS-PAGE and Coomassie blue staining of serum of three homozygous and wild type hens and observed that albumin was also secreted into blood in homozygous animals (**Supplementary Figure S2, Line 146-149 in the revised manuscript**).

Reviewer's comments : How pure are the preparations of rhFc? Is there any chicken IgY or albumin-Fc? The Coomassie gel on Fig 3a looks like the purity is high but the mass spec data in S2 and S3 are confusing. Tables in S2 and S3 should be described, there is no legend.

Author's response : We have purified hIgG1 Fc using protein A column and size exclusion chromatography and confirm purification of hIgG1 Fc in SDS-PAGE. Although the exact purity of sample can be determined by HPLC, we didn't further analysis because it was identified in SDS-PAGE that purity is high enough for conducting further experiments. Additionally, because protein A does not bind to chicken IgY (*Ansari & Chang., Am J Vet Res, 1983; W.W. Zhang., Drug Discov Today, 2003*), we carefully expect that IgY may not be included in our purified hIgG1 Fc samples. In LC/MS/MS, we crop the band of rhIgG1 Fc in SDS-PAGE gel and extract protein, degrade into several peptides, and conduct mass spectrometry. Because degraded peptide fragments are analyzed, not a full protein sequence, there is possibility to annotate unrelated proteins. Although we identified that the annotated proteins with high coverage have amino acid sequences of hIgG1 Fc (NCBI Accession number Q6N096, Q86TT2, A0A286YHEY4, Q6MZX7), the name of proteins associated with accession number are differed from hIgG1 Fc, which may bring confusion to readers. Therefore, for reducing confusions, we suggest that it is better to remove supplementary tables on the LC/MS/MS and we have revised our manuscript. We hope our revision could be acceptable for you.

Reviewer's comments : What is the half-life of the recombinant hFc in mice?

Author's response : According to your comment, we have injected rhIgG1 Fc derived from yolks and recombinant Fc produced from HEK293 cells (Cat No. 10702-HNAH, SinoBiological) into C57BL/6 female mouse and measure serum concentrations of rhIgG1 Fc during several days after injection for measuring half-life. The half-life of rhIgG1 Fc derived from yolks and recombinant Fc produced from HEK293 cells was measured as 39.14 and 36.37 hours, respectively. Please see **Supplementary Fig S3** and **L220-223 in the revised manuscript**.

Reviewer's comments : Could you comment on why such a vast excess (1000-fold) of IVIG or Fc is needed to block ADCC activity or anti-platelet activity in the in vivo model?

Author's response : IVIG is polyclonal IgG antibodies purified from pooled human plasma of thousands of people's blood. Initially, IVIG was used to confer passive immunity to patients with compromised immunity. Meanwhile, it was discovered that when IVIG was administered at high dose (1-2g/kg), it induced anti-inflammatory response and restore platelet counts in ITP patients (*Imbach et al., Lancet, 1981*). Thereafter, high dose IVIG treatments have been widely used in treatment of various inflammatory and autoimmune diseases. Although the exact mechanisms of anti-inflammatory activity of high-dose IVIG have not been clearly elucidated yet, there are several suggested mechanisms such as sialylated Fc mediated anti-inflammatory activities and inhibition of autoantibody-antigen immune complexes binding to Fc receptors by competitive blockade of Fc receptors. Because only 10% of IgG have sialylated Fc in IVIG, it was suggested that vast excess of IVIG should be used to induce anti-inflammatory activities and several studies have shown that enrichment of sialylated IgG can induce anti-inflammatory activities at lower doses (*Kaneko et al., Science, 2006*). In the aspect of Fc receptor blockade, monovalent binding affinity of IgG to Fc γ RII and Fc γ RIII is very low. Therefore, vast excess of IVIG or Fc is required to competitively block binding of auto-immune complexes to these Fc receptors and induce anti-inflammatory activities (*Nagelkerke & Kuijpers., Front Immunol, 2015*). We hope our explanation could satisfy you.

Reviewer's comments : Figure 5d seems to be missing some labeling. The scatter plots should be labeled with what sample is analyzed on each plot.

Author's response : Thank you for your indication of our mistake. We add missing label on scatter plot of **Figure 5d**.

Reviewer's comments : Figure S2. There needs to be a positive control for DC-SIGN binding, otherwise the two negative results from IVIG and Fc are not meaningful. Especially because this was an unexpected result.

Author's response : Thank you for your comment. Although it has been suggested that DC-SIGN can bind to sialylated Fc region of IgG, demonstrated by cell-based ELISA method

(Anthony et al., PNAS, 2008. doi: 10.1073/pnas.0810163105, Sondermann et al., PNAS, 2013. doi: 10.1073/pnas.1307864110), several other reports using FACS or SPR method showed that DC-SIGN did not bind to Fc region of IgG *(Temming et al., Sci Rep, 2019. doi: 10.1038/s41598-019-46484-2, Zhang et al., JCI Insight, 2019. doi: 10.1172/jci.insight.121905)*. Although we don't have positive control for DC-SIGN binding as you pointed, we used SPR method to measure affinity of Fc to several Fc receptors including DC-SIGN and we expect that our data could be interpreted as in accordance with above two reports. We hope our explanation could satisfy you.

Response to Reviewer #2

Reviewer's comments : Park et al. address the point of IVIG being expensive and of limited supply, by suggesting an alternative IgG1 Fc alternative. This was generated using genome-edited chickens, with CRISPR/Cas9-NHEJ, that produce recombinant human IgG-Fc (rhIgG1) in serum and egg yolk. In addition, they show, with LC/MC/MC, that these chickens produce rhIgG1 with high sialylation and low fucosylation levels, features that they show to exert anti-inflammatory effects in an in vitro ADCC assay and a passive ITP mouse model.

The manuscript is written in a concise manner and the data is presented in a logical order. However, the message of the manuscript is oversold and more evidence is needed to support certain statements. The following points of concern can be raised regarding the scientific content of the manuscript

Author's response : The authors are thankful to this reviewer for provided a positive evaluation on our manuscript, and several comments that greatly improved the manuscript. The manuscript was revised extensively and also prepared a point-by-point response to your comments. Therefore, we believe that the revised manuscript could satisfy your point of view on our manuscript. Please see below for our responses on your specific comments, and the manuscript for the corresponding revision.

1. Major Points

Reviewer's comments : NHEJ donor plasmid: For the CRISPR/Cas9-NHEJ design, a T2A tag was used in the donor plasmid in order to separate the ALB protein from the rhIgG1 Fc, but the function, rational and potential limitations of this tag is not explained. E.g. why is it not fully cleaved?

Author's response : To prevent side effects that can be caused by albumin deficient, we linked rhIgG1 Fc to albumin using T2A self-cleavage peptide and we intended to separate rhIgG1 Fc from albumin, although cleavage efficiency of 2A peptides cannot be reached to 100%. We describe the rational of using T2A tag in **Line 125-130 in the revised manuscript** according

to your comments.

Reviewer's comments : It is stated several times that that production of rhIgG1 Fc in chickens can be an alternative source of IVIG that reduces the costs. This seems to be unlikely approach towards FDA approved product, and that is not taking into account that the concentration in blood and yolk is rather low, making this non-feasible.

Author's response : Thank you for your comment to improve quality of our manuscript. During preparation of manuscript, we intended to suggest that chicken liver specific expression system can produce recombinant proteins with higher sialylation and lower fucosylation ratio, and this glycosylation patterns could be beneficial to anti-inflammatory activity of IgG1 Fc. Based on this idea, we think chicken can be one of potential alternative source of human IVIG as anti-inflammatory agents. In present study, IgG1 Fc can be accumulated as 3-4 mg/egg and because chicken eggs can be produced with cost-effectiveness (around 10 cents per egg), we suggested here that chicken bioreactor can be cost-effective system for producing human IgG1 Fc. However, we agreed to your points that more improved efficiency of human Fc production in eggs should be required to solve the problem of supply of IVIG and we will continuously research to improve efficiency of our system. According to your comment, we have revised manuscript in general to reduce our description on the cost-effectiveness of our system (**Line 30-31, Line 114, Line 308-309 in the revised manuscript**) and we changed title of manuscript "*Human IVIG alternative with beneficial N-glycosylation pattern for anti-inflammatory activity derived from genome edited chickens*" to "*Production of recombinant human IgG1 Fc with beneficial N-glycosylation pattern for anti-inflammatory activity using genome edited chickens*".

Reviewer's comments : The anti-inflammatory effects of sialylated IgG-Fc and also DC-SIGN as receptor for IgG are highly debatable and been a topic of a body of literature suggesting previous papers claiming DC-SIGN to be a receptor for human IgG to be faulty. Results you present and discuss in line 225 are in accordance with that recently published by Temming et al Scientific reports 9.1 (2019): 1-10) suggesting DC-SIGN does not bind human IgG. This needs to be mentioned.

Author's response : Thank you for your comments. We have mentioned above reference about the DC-SIGN was not bona fide receptor for human IgG1 Fc, as you recommended in discussion section **Line 350-356 in the revised manuscript**.

Reviewer's comments : L268. The in vivo mouse model is not introduced in the result section. Explain the model and cite an original reference for this (e.g. Blood. 2001;98(4):1095-1099).

Author's response : According to your comment, we introduce in vivo mouse model and cite original reference for this model as you recommended (**Line 279-281 in the revised**

manuscript).

Reviewer's comments : L156 and L159 and Fig2b. In the text, a band of 50kDa is mentioned and this is also observed in the figure. However, the text next to the band in the figure states 70kDa. Moreover, the reducing band is 35, and the authors explain that this is the size of glycosylated CH₂+CH₃. But is the non-reduced 50kDa band then deglycosylated?

Author's response : As you pointed out, we found that our description is rather confusing to readers. We have not performed any deglycosylation in our western blot experiment in Figure 2b. In reducing conditions, disulfide bonds are cleaved and the proteins are linearized and localized to its exact molecular weight band size when SDS-PAGE gel running. However, in non-reducing conditions, the proteins still folded into 3D structure and because of this nature, the band size of target proteins tends to be down-shifted than expected molecular weight. To prevent confusion, we deleted band size description in non-reducing condition in manuscript (**Line 160-161 in the revised manuscript**) and in **Figure 2B**.

Reviewer's comments : The authors performed an in vitro ADCC experiments with FcγRIIIa-expressing Jurkat cells. Jurkat cells are T cell-derived that do not express FcγR (<https://www.ncbi.nlm.nih.gov/pmc/articles/PMC3994145/>). Where did the FcγRIII originate, were these cells generate bought or were these WT Jurkat?

Author's response : In ADCC experiments, we used ADCC Reporter Bioassay, V variant complete kit (WIL2-S) manufactured by Promega (Cat No. G7014). The transgenic Jurkat cells that express human FcγRIIIA (V158) was provided in this kit and used as effector cells. These transgenic Jurkat cells also have been engineered to induce luciferase activity when NFAT pathway is activated by FcγRIIIA crosslinking. We have incubated these effector cells with serially diluted anti-CD20 antibody, WIL2-S target cells that express CD20, and IVIG or rhIgG1 Fc and examine the level of ADCC induction (**Line 250-256 and Methods section (Line 532-541) in the revised manuscript**).

Reviewer's comments : The effector cells used for the ADCP assay are not described nor the principles of the assay. Its also very surprising that IVIg nor the Fc fragment has an effect. This has been described by numerous other papers.

Author's response : In ADCP assay, we used FcγRIIA-H ADCP Bioassay kit manufactured by Promega (Cat No. G9901) and assay principles are same with ADCC assay. We have described ADCP measuring principles more detail in **Line 268-276**. Although ADCC was effectively inhibited in 1.8 mg/ml of IVIG and 0.6 mg/ml of rhIgG1 Fc, ADCP was not effectively inhibited in same concentration of IVIG and rhIgG1 Fc. This may be resulted from relatively lower affinity of IVIG and rhIgG1 Fc to FcγRIIA than FcγRIIIA and more higher concentration of IVIG and rhIgG1 Fc will be required to have blocking ability to FcγRIIA in

our experimental setting. We have changed our description as FcγRIIA blocking activity cannot be observed at concentration of 1.8 mg/ml of IVIG and 0.6 mg/ml of Fc (**Line 268-276 in the revised manuscript**). Also, we describe assay method more specifically in **Materials and methods section**. We hope our explanation could be acceptable for your standard.

Reviewer's comments : The main text is often written in large paragraphs. Please try to be a little more concise and split up those large paragraphs in smaller logical units to improve readability.

Author's response : We appreciate for your comment to improve quality of our manuscript. We revised large paragraphs into more smaller units (**Line 61-62, 76-77, 130-131, 142-143, 157-158, 169-170 in the revised manuscript**) according to your recommendation.

2. Minor Points

Reviewer's comments: Explain a little better in the introduction that afucosylation of antigen-specific IgG is proinflammatory, but bulk antigen-specific IgG can be anti-inflammatory due to blocking of FcγRIIIa. Two papers actually have shown that FcγRIIIa are preferentially occupied by aspecific afucosylated IgG1 in humans in vivo. (<https://pubmed.ncbi.nlm.nih.gov/31748349/> and DOI:10.1074/mcp.RA119.001607).

Author's response: According to your comments, we further explain afucosylated antigen-specific antibody can be anti-inflammatory and cited references of your recommendation (**Line 72-76 in the revised manuscript**).

Reviewer's comments: L30. Spell out ITP at first appearance as immune thrombocytopenia

Author's response: We spell out ITP as immune thrombocytopenia according to your recommendation.

Reviewer's comments: T94. 'do not produce non-human glycans'. Difficult sentence, could be changed into 'only produce human glycans' or something similar.

Author's response: We have changed our description according to your recommendation (**Line 90 in the revised manuscript**).

Reviewer's comments: L97. Difficult sentence, write down differently.

Author's response: We have changed our description for improving readability according to your recommendation (**Line 92-94 in the revised manuscript**)

Reviewer's comments: L113. Write rhIgG1 directly after 'The recombinant human IgG1 Fc.....'

Author's response: We corrected main text according to your recommendation (**Line 110**).

Reviewer's comments: L115. Change 'chicken' to 'chickens'. This also applies for L121.

Author's response: We corrected words according to your recommendation (**Line 113 and Line 118**).

Reviewer's comments: L138. Fig. 1e is written there but should be removed.

Author's response: We corrected text according to your recommendation (**Line 140**)

Reviewer's comments: L145. Explain why G1, G2 and G3 progeny are needed. As heterozygous G2 is mated resulting in homozygous G3, the question is whether G3 is also fertile? Is there also reduced albumin expression in G2 and G3 and does this have consequences? Moreover, in the next experiments, are the G2 or G3 used? This is unclear from the text.

Author's response: In G1, we obtained only one *ALB::hIgG1 Fc* rooster that finally reached sexual maturation. Therefore, we perform mating between G1 *ALB::hIgG1 Fc* rooster and wild type hen, producing G2 heterozygous *ALB::hIgG1 Fc* progenies. By mating between G2 progenies, we produced G3 progenies with homozygous for *ALB::hIgG1*. The homozygous *ALB::hIgG1* chickens can reach sexual maturation and lay eggs, which means homozygous *ALB::hIgG1* chickens are also fertile. rhIgG1 Fc also accumulated in egg yolk of eggs laid by homozygous *ALB::hIgG1* chickens. We add data on the analysis of serum and egg yolk of homozygous *ALB::hIgG1* chickens in **Supplementary Figure S2**. And also please see **L146-150**. We hope our revision could satisfy you.

Reviewer's comments: L153. 'expressed' needs to be 'transcribed' as we are talking about RNA.

Author's response: We corrected 'expressed' into 'transcribed' as your recommendation. Please see **L157**.

Reviewer's comments: L154. ALB::hIgG1 Fc is ONLY transcribed in the liver. This is not the organ expressing the highest amount, as the other organs don't show the presence of ALB::hIgG1 Fc RNA.

Author's response: We changed our description as "rhIgG1 Fc was transcribed successfully in the liver specific manner" (**Line 156-157**)

Reviewer's comments: L168. An error bar of +/- 71,65 is mentioned in the text, but the error bar in the corresponding figure (Fig 2c) is very small.

Author's response: We corrected our errors. Please see **Fig 2c**.

Reviewer's comments: L175. 'generation' is confusing, rather use progenies or something similar.

Author's response: We corrected main text as your recommendation and change "generation" to "progenies". Please see **L177**.

Reviewer's comments: L194. A comparison is made between ALB::hIgG1 Fc glycosylation and human glycosylation patterns, but this is not clearly explained. In addition, use proper reference and mention the IgG glycosylation profile observed in human serum.

Author's response: We revised our manuscript according to your comments with reference describing N-glycosylation pattern of human serum proteins. Please see **L372-377 of revised manuscript**.

Reviewer's comments: L205. Add the word 'the' in 'is largely THE same as'. This also applies for L207 'to identify THE major linkage'.

Author's response: We added 'the' according to your recommendation. Please see **L209 and L211**.

Reviewer's comments: L233. As this result section is based on SPR data, the conclusion that rhIgG1 Fc has higher affinity for FcγRIIIa can be made, but the conclusion that it also has higher blocking activity does not fit here.

Author's response: We revised manuscript according to your comment. Please see **L240-241**.

Reviewer's comments: L277. Anti-inflammatory activity is mentioned here, but this is rather blocking activity. It would be more clear if these two definitions are more strictly separated in the last two result sections.

Author's response: We changed our description according to your recommendation. Please see **L279-291**

Reviewer's comments: L307. The word efficient is mentioned two times.

Author's response: We revised 'efficient sialylation efficiency' to 'efficient sialylation ratio' Please see **L317**.

Reviewer's comments: L310 and L313. Try to find a different word for 'also' in one of the sentences. This also applies for L355 and L357 where the word suggested is used twice.

Author's response: We changed 'Also' into 'Additionally' in **L323** and changed 'suggested' into 'proposed' in **L372**.

Reviewer's comments: L357. Make this message more clear. What is meant by human blood products? IgG or also other proteins?

Author's response: In this statement, we intended to suggest that although it is not fully demonstrated yet, chicken liver bioreactor can be one of optimal production platform for human blood products that are synthesized from human liver such as blood clotting factors and alpha-1 antitrypsin because glycosylation pattern of liver derived proteins from these two species is similar. We tried to clarify the message by revising our description at **L372-377** according to your recommendation.

Reviewer's comments: Fig1b. Explain DW

Author's response: We changed term DW (distilled water) to ddH₂O. Please see Fig 1b.

Reviewer's comments: L596. Add the word 'for' after 'using primers specific FOR...'

Author's response: We corrected legend of Figure 1 according to your comments.

Reviewer's comments: Fig3e. It would be nice to have a positive control for the MAL II blot.

Author's response: According to your recommendation, we used recombinant EPO derived from CHO cells (Cat No. 100-64, Peprotech) as positive control for the MAL II blot and revised figure. Please See Fig3e of the revised manuscript.

Reviewer's comments: Fig5d. Mention PBS, IVIG and rhIgG1 conditions in the FACS dotplots as they are lacking.

Author's response: Thank you for your indication of our mistake. We add missing label on scatter plot of Figure 5d.

Reviewer's comments: Supplementary figure 2: the scale and overall layout for the Fc and binding do Fc γ R2 seems off– all the points seem on the Y axes at 0 M (Is the unit of the X axes correct?) with a random line protruding from the X at ca 3.3 M.

Author's response: As you mentioned, in our SPR experiment, we cannot detect any binding between rhIgG1 Fc and Fc γ R2 in all range of concentration. The values plotted on the sensorgram and units for both X and Y axis are automatically analyzed by BiaEvaluation 3.01 software and we used sensorgram without any modifications.

Reviewer's comments: Eggs, normally containing rather large amounts of IgY antibodies, are consumed. It would be interesting to see in future projects how well the Fc fragments survive the gastro enteric tract and if they are taken up by FcRn in the gut, starting with mouse models.

Author's response: Thank you for your suggestion on the future research project and give us opportunity for studying FcRn. Based on your suggestion, we will continue to research on the delivery of yolk Fc into gastro enteric tract or nasal cavity to apply development of edible vaccines or bio-drugs formulated by egg yolk.

Revision #2

Response to Reviewer #1

We appreciate to Reviewer #1 for providing detailed review and valuable comments that help us to improve quality of our manuscript. Based on your comments, we revised our manuscript and we hope that our revisions could be acceptable for you.

Reviewer's comment : Typo in the new title (in the PDF version): ...“using from genome edited chickens” should be changed to either “using” or “from”.

Author's response : We revised the title as ...“using genome edited chickens”. Please see Line 2.

Reviewer's comment : Lines 139, 141, 177, 435: "progeny" is both singular and plural (i.e. don't use “progenies”)

Author's response : We corrected words. Please see Lines 139, 141, 178, 435.

Reviewer's comment : Line 177, not sure what is meant here by progeny. Generation? Zygoty? Genetic lineage from original PGCs?

Author's response : In here, we intended to describe that Fc could be continuously secreted into bloodstream during several generations without transgene silencing. We revised description for more clear delivery as “regardless of progeny in several generations” Please see Line 178.

Reviewer's comment : Mendelian inheritance should be confirmed in the breeding to homozygosity (i.e. 1:2:1 ratio of genotypes in progeny of heterozygous matings), in S1.

Author's response : According to your comments, we have analyzed genotype of chicks hatched from heterozygous mating. From analyzed 18 chicks, we obtained four wild type, nine heterozygous and five homozygous chicks, which approximately follow 1:2:1 ratio of Mendelian inheritance. Please see Figure S1. and Line 146-148.

Reviewer's comment : The chimera test mating needs to be better explained. According to the legend, the transplanted donor PGCs are I/I, the KO recipients of the PGCs are I/i and the wild type hens used for mating to the chimeras are I/I. Thus both donor-derived and recipient-derived progeny could be I/I, and half of the recipient-derived progeny would be I/i. Thus to calculate frequency of germline transmission you are looking for loss of I/i, not gain of a specific genotype, which makes it much less useful than if a specific genotype is associated

with germline transmission.

Author's response : We found error in Table S1 legend. The genotype of KO recipient is *i/i*, not *I/i*. We used PGCs derived from WL (*I/I*) and transplanted these PGCs into KO (*i/i*) recipients. Therefore, germline chimeric KO could produce sperm of *I* (derived from WL donor PGCs) and *i* (derived from KO endogenous germ cell). After mating between wild type WL (*I/I*) hen and germline chimeric KO, the donor PGC derived progeny will be *I/I* (WL) and endogenous KO germ cell derived progeny will be *I/i* (hybrid). In this regards, for calculating germline transmission efficiency, we calculated the ratio of donor PGC derived progeny (*I/I*) from total hatched chicks. We have revised legend of Table S1.

Reviewer's comment : Figure S2: calculation of ALB concentration in serum and/or eggs from WT, het and hom birds would be much more informative than the Coomassie gel only showing WT and homozygous

Author's response : According to your comments, we have calculated ALB concentration of WT, heterozygous and homozygous hens. The ALB protein secreted into bloodstream regardless of genotype although its concentration have a tendency to decrease in heterozygous and homozygous birds compared to wild type birds. Please See Figure S2 and Line 148-150.

Reviewer's comment : Line 147: sentence is not right: “We observed that homozygous chickens also secrete ALB into blood and healthy to have sexual maturation, and lay eggs” should be changed to something like “We observed that homozygous chickens also secrete ALB into blood, are healthy, reach sexual maturity, and lay eggs”

Author's response : We revised sentence according to your comments and please see Line 148-150.

Reviewer's comment : Line 151: “maintained as a homozygous breed” has not been shown; so far, you have shown that eggs are laid by homozygous females but not that they would produce viable offspring. You either need to show hatching and rearing of chicks from homozygous parents, or remove the statement.

Author's response : We agreed to your comment and removed related statement. Please see Line 151-152.